# Thymic macrophages consist of two populations with distinct localization and origin

Tyng-An Zhou[1], Hsuan-Po Hsu[1], Yueh-Hua Tu[2,3], Hui-Kuei Cheng[1], Chih-Yu Lin[1], Nien-Jung Chen[1], Jin-Wu Tsai[4], Ellen A Robey[5], Hsuan-Cheng Huang[2,6], Chia-Lin Hsu[1], Ivan L Dzhagalov[1]*

[1]Institute of Microbiology and Immunology, National Yang Ming Chiao Tung University, Taipei, Taiwan; [2]Bioinformatics Program, Taiwan International Graduate Program, Institute of Information Science, Academia Sinica, Taipei, Taiwan; [3]Graduate Institute of Biomedical Electronics and Bioinformatics, National Taiwan University, Taipei, Taiwan; [4]Brain Research Center, National Yang Ming Chiao Tung University, Taipei, Taiwan; [5]Division of Immunology and Pathogenesis, Department of Molecular and Cell Biology, University of California, Berkeley, Berkeley, United States; [6]Institute of Biomedical Informatics, National Yang Ming Chiao Tung University, Taipei, Taiwan

**Abstract** Tissue-resident macrophages are essential to protect from pathogen invasion and maintain organ homeostasis. The ability of thymic macrophages to engulf apoptotic thymocytes is well appreciated, but little is known about their ontogeny, maintenance, and diversity. Here, we characterized the surface phenotype and transcriptional profile of these cells and defined their expression signature. Thymic macrophages were most closely related to spleen red pulp macrophages and Kupffer cells and shared the expression of the transcription factor (TF) SpiC with these cells. Single-cell RNA sequencing (scRNA-Seq) showed that the macrophages in the adult thymus are composed of two populations distinguished by the expression of *Timd4* and *Cx3cr1*. Remarkably, *Timd4*+ cells were located in the cortex, while *Cx3cr1*+ macrophages were restricted to the medulla and the cortico-medullary junction. Using shield chimeras, transplantation of embryonic thymuses, and genetic fate mapping, we found that the two populations have distinct origins. *Timd4*+ thymic macrophages are of embryonic origin, while *Cx3cr1*+ macrophages are derived from adult hematopoietic stem cells. Aging has a profound effect on the macrophages in the thymus. *Timd4*+ cells underwent gradual attrition, while *Cx3cr1*+ cells slowly accumulated with age and, in older mice, were the dominant macrophage population in the thymus. Altogether, our work defines the phenotype, origin, and diversity of thymic macrophages.

## Editor's evaluation

This work provides thorough characterization of thymic macrophages. The authors used bulk RNA-seq, single-cell RNA-seq and fate mapping animal models to demonstrate the phenotypes, origin and diversity of thymic macrophages. The manuscript is well written and the conclusions are mostly well supported by data.

## Introduction

Tissue-resident macrophages are present in every organ and maintain local homeostasis through diverse functions ranging from protection against pathogens to tissue repair (*Wynn et al., 2013*).

**\*For correspondence:**
ivan.dzhagalov@nycu.edu.tw

**Competing interest:** The authors declare that no competing interests exist.

To perform their roles efficiently, macrophages acquire specialized phenotypes depending on the tissue microenvironment, and as a consequence, multiple subtypes exist, frequently within the same organ. For example, the spleen harbors red pulp macrophages specialized in red blood cell phagocytosis, marginal zone macrophages, and metallophilic macrophages that are the first defense against blood-borne pathogens, T cell zone macrophages that silently dispose of apoptotic immune cells, and tingible-body macrophages that engulf less fit B cells in the germinal center (*Baratin et al., 2017*; *A-Gonzalez and Castrillo, 2018*; *Bellomo et al., 2018*). Thus, tissue-resident macrophages represent a fascinating developmental system that allows enormous plasticity.

The last decade has seen a paradigm shift in our understanding of the development of tissue-resident macrophages. Contrary to the long-held belief that all macrophages derive from circulating monocytes (*van Furth and Cohn, 1968*), multiple studies have shown that many of them are long-lived cells with an embryonic origin that can maintain themselves in the tissues (reviewed in *Ginhoux and Guilliams, 2016*). Three waves of distinct progenitors settle the tissues and contribute in various degrees to the resident macrophages in each organ. The first wave consists of the yolk sac (YS)-derived primitive macrophages that enter all tissues and establish the earliest macrophage populations (*Perdiguero and Geissmann, 2016*; *Mass et al., 2016*). In all organs, except for the brain and, partially, the epidermis, primitive macrophages are replaced by the next wave consisting of fetal monocytes (*Ginhoux et al., 2010*; *Hoeffel et al., 2012*; *Hoeffel et al., 2015*; *Goldmann et al., 2016*). The third wave comes from hematopoietic stem cells (HSCs)-derived monocytes that contribute to various degrees to the macrophage pool in different tissues. For example, these cells contribute little to the microglia in the brain, Langerhans cells in the epidermis, and alveolar macrophages in the lungs but substantially to most other organs (*Hashimoto et al., 2013*; *Epelman et al., 2014*; *Sheng et al., 2015*; *Liu et al., 2019*). Moreover, the kinetics and timing of HSC-derived monocyte infiltration vary in different parts of the body. For some macrophage populations, such as the arterial macrophages and subcapsular lymph node macrophages, monocytes replace embryonic macrophages soon after birth and self-maintain after that with little contribution from circulating cells (*Ensan et al., 2016*; *Mondor et al., 2019*). Others, such as heart macrophages, osteoclasts, and pancreatic islets macrophages, are progressively replaced at a low rate (*Epelman et al., 2014*; *Molawi et al., 2014*; *Heidt et al., 2014*; *Calderon et al., 2015*; *Jacome-Galarza et al., 2019*; *Yahara et al., 2020*). A third group, such as the macrophages in the dermis and most of the gut macrophages, is constantly replaced by blood monocytes with relatively fast kinetics (*Tamoutounour et al., 2013*; *Bain et al., 2014*). These conclusions have been extended to many different macrophage populations such as Kupffer cells, liver capsular macrophages, red pulp macrophages, testicular macrophages, large and small peritoneal macrophages, and T zone macrophages in the lymph nodes (*Baratin et al., 2017*; *Hashimoto et al., 2013*; *Epelman et al., 2014*; *Liu et al., 2019*; *Sierro et al., 2017*; *Mossadegh-Keller et al., 2017*; *Lokka et al., 2020*; *Wang et al., 2021*; *Bain et al., 2016*).

The recent revitalization in macrophage research has yet to reach thymic macrophages. Although their prodigious phagocytic ability is well appreciated (*Surh and Sprent, 1994*), little is known about the origin, diversity, and maintenance of these cells. This gap in our knowledge is, partly, due to the lack of a consensus about the surface phenotype of thymic macrophages. Various groups have used different markers to identify these cells, such as F4/80 and Mac-3 (LAMP-2) (*Surh and Sprent, 1994*), or CD4 and CD11b (*Esashi et al., 2003*), or Mac-2 (galectin 3), F4/80, and ED-1 (CD68) (*Liu et al., 2013*). Most commonly, researchers employ F4/80 and CD11b (*Guerri et al., 2013*; *Lopes et al., 2018*; *Kim et al., 2010*; *Tacke et al., 2015*). However, none of these markers is macrophage-specific: F4/80 is also expressed on eosinophils and monocytes (*Gautier et al., 2012*; *Ingersoll et al., 2010*), while CD11b is present on most myeloid cells. The lack of a clear phenotypic definition of thymic macrophages has translated into the absence of models that targets genes specifically in this population. For example, although macrophages in various organs have been successfully targeted with *Lyz2^Cre^*, *Csf1r^Cre^*, or *Cx3cr1^Cre^*, very few studies have used these models in the thymus (*Tacke et al., 2015*; *Wang et al., 2019*; *Chan et al., 2020*).

Only a handful of studies have explored the origin of thymic macrophages. Several reports have indicated that these cells could be derived from T cell progenitors in the thymus based on an improved single-cell in vitro culture and in vivo transplantation experiments (*Wada et al., 2008*; *Bell and Bhandoola, 2008*). However, these conclusions have been questioned based on fate-mapping experiments using *Il7r^Cre^* that found very limited contribution of lymphoid progenitors to thymic macrophages in

vivo in unperturbed mice (*Schlenner et al., 2010*). Most recently, *Tacke et al., 2015* used parabiosis to rule out circulating monocytes as a major source of thymic macrophages. The same study also performed fate-mapping experiments to show that most thymic macrophages descend from *Flt3*+ HSC-derived progenitors. However, the contribution of earlier waves of hematopoiesis has not been explored.

Here, we aimed to bring our knowledge of thymic macrophages on par with other tissue-resident macrophages. We started by clearly defining thymic macrophages according to the guidelines set by the Immunological Genome Consortium (IMMGEN) (*Gautier et al., 2012*) and characterized their surface phenotype and transcriptional signature. Using scRNA-Seq, we identified two populations of thymic macrophages with distinct localization. We explored the origin of these cells through genetic fate mapping, shield chimeras, and embryonic thymus transplantation and documented that different waves of progenitors give rise to the two populations of thymic macrophages. Altogether our work fills an important gap in our understanding of resident thymic macrophages and provides the framework for future functional characterization of these cells.

## Results

### CD64, F4/80, and MerTK identify the macrophages in the thymus

To unambiguously and comprehensively identify macrophages in the thymus, we evaluated several of the prototypical macrophage markers – MerTK, CD64, and F4/80 (*Gautier et al., 2012*) – a population that was stained with all three markers (*Figure 1A*). As staining with MerTK and F4/80 was relatively dim even when the brightest fluorochromes (e.g. PE) were used and could not be resolved fully from the isotype control (*Figure 1—figure supplement 1*), we chose to use CD64 vs. forward scatter (FSC) as the first step in our gating strategy (*Figure 1B*). Among CD64+FSC$^{hi}$ cells, F4/80+CD11b$^{lo}$ macrophages could be distinguished from F4/80$^{lo}$CD11b+ monocytes.

The CD64+F4/80+MerTK+CD11b$^{lo}$FSC$^{hi}$ cells had typical macrophage morphology with abundant cytoplasm (*Figure 1C*). These cells did not express lineage markers characteristic of T cells (CD3ε), B cells (CD19), eosinophils (Siglec F), NK cells (NK1.1), neutrophils (Gr1), or plasmacytoid dendritic cells (Siglec H) (*Figure 1D*). However, they expressed phagocytic receptors such as TIM4, CD51, and Axl (*Figure 1E*). Immunofluorescent staining with CD64, MerTK, and TIM4 in the thymic cortex confirmed the presence of large cells positive for all three macrophage markers (*Figure 1—figure supplement 2*).

Importantly, MerTK+ cells could not be labeled by intravenously injected CD45 antibody (*Figure 1F*), proving that they reside in the parenchyma of the organs and not in the blood vessels. Based on the above data, we will refer to CD64+F4/80+MerTK+CD11b$^{lo}$FSC$^{hi}$ cells as thymic macrophages. The smaller CD64+F4/80$^{lo}$CD11b+FSC$^{hi}$ population did not express MerTK but most of them expressed Ly6C, and we classified them as thymic monocytes.

Thymic macrophages expressed CD11c, MHC2, and SIRPα making them partially overlap with CD11c+MHC2+ classical dendritic cells (cDCs), thus making problematic the unambiguous identification of thymic cDCs based only on these two markers (*Figure 1—figure supplement 3*). Proper identification of cDC in the thymus requires the exclusion of macrophages based on CD64 or MerTK staining. Otherwise, the cDCs, particularly the SIRPα+ cDC2 subset, would be contaminated with macrophages that account for ~25% of cDC2 (*Figure 1—figure supplement 4*).

Thymic macrophages were ~0.1% of all the cells in the thymus of young adult mice and numbered ~4×10⁵ on average per mouse (*Figure 1G*). We did not find statistically significant differences in their percentages between 4 and 11 weeks of age. Still, there was a significant decline in their numbers with age, consistent with the beginning of thymic involution (*Figure 1H*).

### Transcriptional signature of thymic macrophages

To further understand the identity and functions of the thymic macrophages, we analyzed the RNA sequencing data from the IMMGEN's Open Source Mononuclear Phagocyte profiling. We first examined the expression of the core signature macrophage genes (*Gautier et al., 2012*) and found that they were enriched in thymic macrophages but not in *Sirpa*+ or *Xcr1*+ thymic cDCs (*Figure 2A*). On the contrary, cDC core signature genes were abundantly expressed in both thymic cDC subsets but not in

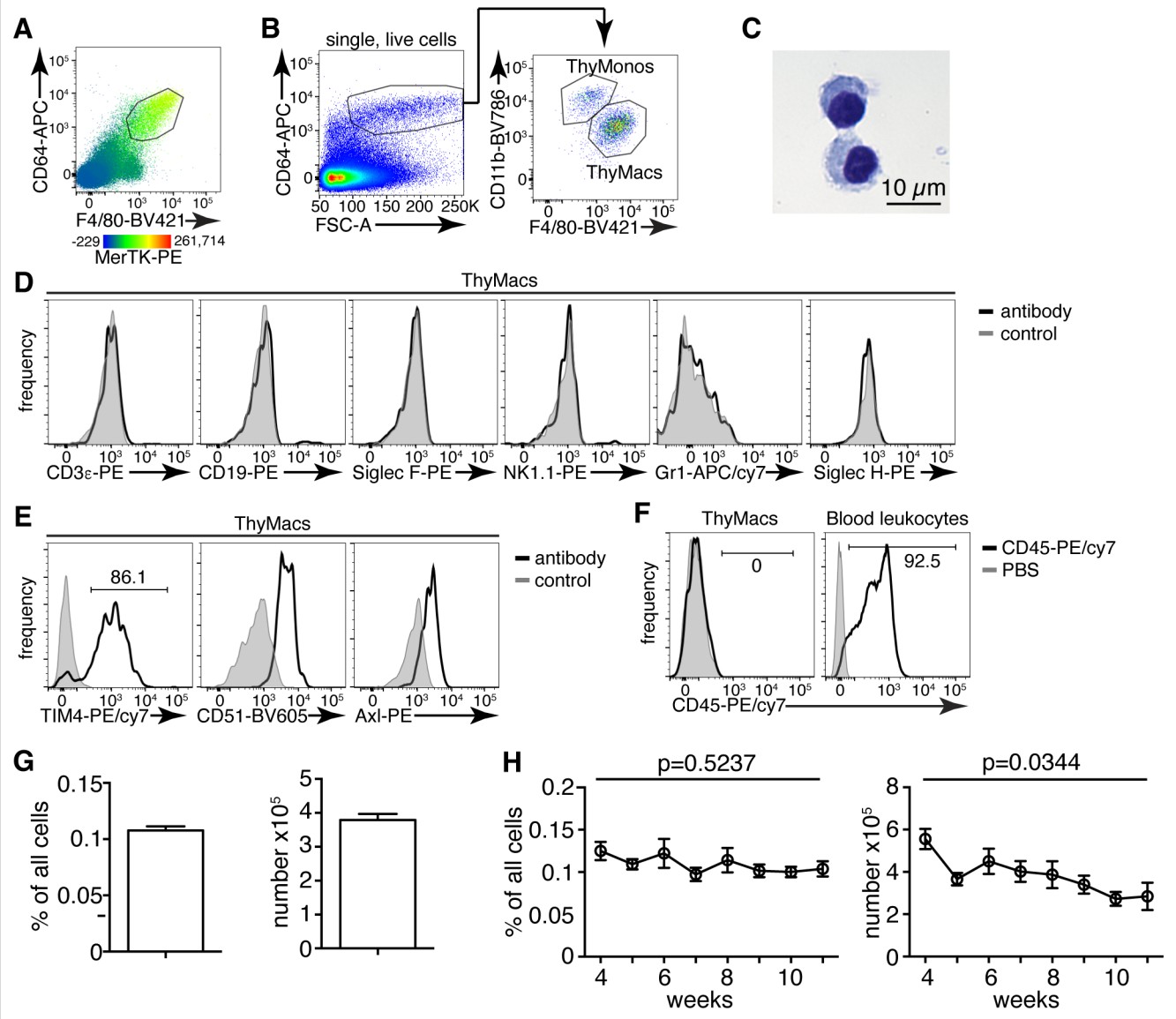

**Figure 1.** Thymic macrophages (ThyMacs) can be identified by the expression of CD64, MerTK, and F4/80. (**A**) Flow cytometric analysis of enzymatically digested thymus tissue with macrophage markers CD64, MerTK, F4/80, and CD11b. (**B**) Gating strategy for identifying ThyMacs: CD64+FSC$^{hi}$ are first gated; the F4/80+CD11b$^{lo}$ cells among them are the ThyMacs, while F4/80$^{lo}$CD11b+ are the thymic monocytes (ThyMonos). (**C**) Pappenheim (Hemacolor Rapid staining kit) staining of sorted ThyMacs. (**D**) Lack of expression of lineage markers associated with other cell types on ThyMacs. (**E**) The expression on ThyMacs of three receptors for phosphatidylserine that participates in the phagocytosis of apoptotic cells. (**F**) Labeling of ThyMacs with intravenously injected anti-CD45-PE antibody or PBS. The labeling of blood leukocytes is shown for comparison. (**G**) Average numbers and percentages of ThyMacs in 4–11 weeks old mice, n=82. (**H**) Comparison of the numbers and percentages of ThyMacs in mice of different ages, n=82. All flow cytometry plots are representative of at least three independent repeats. The numbers in the flow cytometry plots are the percent of cells in the respective gates. Data in (**G**) and (**H**) represent mean ± SEM. Statistical significance in (**H**) was determined with one-way ANOVA.

The online version of this article includes the following source data and figure supplement(s) for figure 1:

**Source data 1.** Numbers and percentages of mouse thymic macrophages.

**Figure supplement 1.** Representative flow cytometry staining of enzymatically digested thymus single-cell suspension for CD64, MerTK, and F4/80 and respective isotype controls.

**Figure supplement 2.** Immunofluorescent images of thymic sections showing co-localization of MerTK and CD64 staining (upper row) and TIM4 and CD64 staining (lower row) in the thymic cortex.

**Figure supplement 3.** Expression of CD11c, MHC2, and SIRPα on thymic macrophages with respective controls.

*Figure 1 continued on next page*

Figure 1 continued

**Figure supplement 4.** Example flow cytometry plots showing that gating on CD11c⁺MHC2⁺ thymus cells, in addition to dendritic cells (DCs), also includes macrophages, especially among SIRPα⁺ cells.

**Figure supplement 4—source data 1.** Thymic macrophages contaminate conventional dendritic cells 2 (cDC2) flow cytometry gate.

thymic macrophages. Thus, although thymic macrophages and cDCs share the thymic microenvironment and expression of CD11c and MHC2, they have distinct transcriptional profiles.

We then compared the gene expression profile of thymic macrophages to that of other well-characterized macrophage populations from the IMMGEN database. Because of the abundance of samples, we limited our comparison to only nine types of tissue-resident macrophages under steady-state conditions – splenic red pulp macrophages, Kupffer cells, broncho-alveolar lavage macrophages, large peritoneal cavity macrophages, white adipose tissue macrophages, aorta macrophages, central nervous system microglia, and spinal cord microglia. Principal component analysis revealed that thymic macrophages were most closely related to splenic red pulp macrophages and Kupffer cells (*Figure 2B*).

To better identify the unique functions of thymic macrophages, we looked for differentially expressed genes in these cells compared to other tissue-resident macrophages. We set three criteria: (1) high expression in thymic macrophages (>500); (2) >fivefold higher expression than the average value in the nine populations of non-thymic macrophages; (3) expression in thymic macrophages is higher than any non-thymic macrophage samples. A total of 44 genes met these criteria, and we consider them to constitute the transcriptional signature of thymic macrophages (*Figure 2C*). These included several degradation enzymes and their inhibitors (*Cst7*, *Mmp2*, *Mmp14*, *Dnase1l3*, *Serpina3g*, *Acp5*), non-classical MHC molecules (*H2-M2*, *H2-Q6*, *H2-Q7*), metabolic enzymes (*Chst2*, *Ass1*, *Kynu*, *Cp*, *Dgat2*, *Sorl1*, *Lap3*), molecules involved in innate immunity (*Ifit2*, *Il18bp*, *Mefv*, *Lgals3bp*), and extracellular signaling molecules and their receptors (*Pdgfa*, *Cxcl16*, *Il2rg*, *Gpr157*). We also looked for TFs highly expressed in thymic macrophages and could potentially regulate critical gene networks in these cells. A total of 25 TFs were highly expressed in thymic macrophages (>250) and were at least twofold higher as compared to the non-thymic macrophages (*Table 1*). Among them were several TFs involved in type I interferon (IFN-I) signaling (*Stat1*, *Stat2*, *Irf7*, and *Irf8*) and lipid metabolism (*Nr1h3*, *Pparg*, *Srebf1*, and *Rxra*) (*Figure 2D*). Notably, *Runx3*, which is essential for the development and function of cytotoxic T lymphocytes (*Taniuchi et al., 2002*), innate lymphoid cells (*Ebihara et al., 2015*), and Langerhans cells (*Fainaru et al., 2004*), was highly expressed in thymic macrophages. *Spic*, which has well-documented roles in the development of red pulp macrophages in the spleen and bone marrow macrophages (*Kohyama et al., 2009*; *Haldar et al., 2014*), was also highly expressed in thymic macrophages, further strengthening the argument for the similarity between thymus, spleen, and liver macrophages. To confirm the expression of *Spic* in thymic macrophages, we analyzed the thymus of *Spic^GFP* mice (*Haldar et al., 2014*). We found that all *Spic^GFP+* cells were macrophages (*Figure 2—figure supplement 1*), making them the most specific thymic macrophage reporter strain compared with *Lyz2^GFP*, MAFIA (*Csf1r^GFP*), *Cd11c^YFP*, and *Cx3cr1^GFP* mice (*Figure 2—figure supplement 2*). However, only ~80% of thymic macrophages were *Spic^GFP+* suggesting heterogeneity within the cells (*Figure 2—figure supplement 3*).

Several dominant pathways emerged when we grouped the 500 most highly expressed genes in thymic macrophages according to gene ontology (GO) terms (*Figure 2E*). Notably, 5 of the 10 most highly enriched GO pathways concerned antigen presentation of both exogenous and endogenous antigens. These data complement our flow cytometry findings of expression of MHC2 and suggest that thymic macrophages could be potent antigen-presenting cells and might play a role in negative selection or agonist selection of thymocytes. Two other highly enriched GO pathways were involved in lysosomal biogenesis and functions, highlighting the high capacity of these cells to degrade phagocytosed material. Thus, our transcriptional analysis has revealed that thymic macrophages are bona fide macrophages that bear significant similarity to spleen and liver macrophages and are specialized in lysosomal degradation of phagocytosed material and antigen presentation.

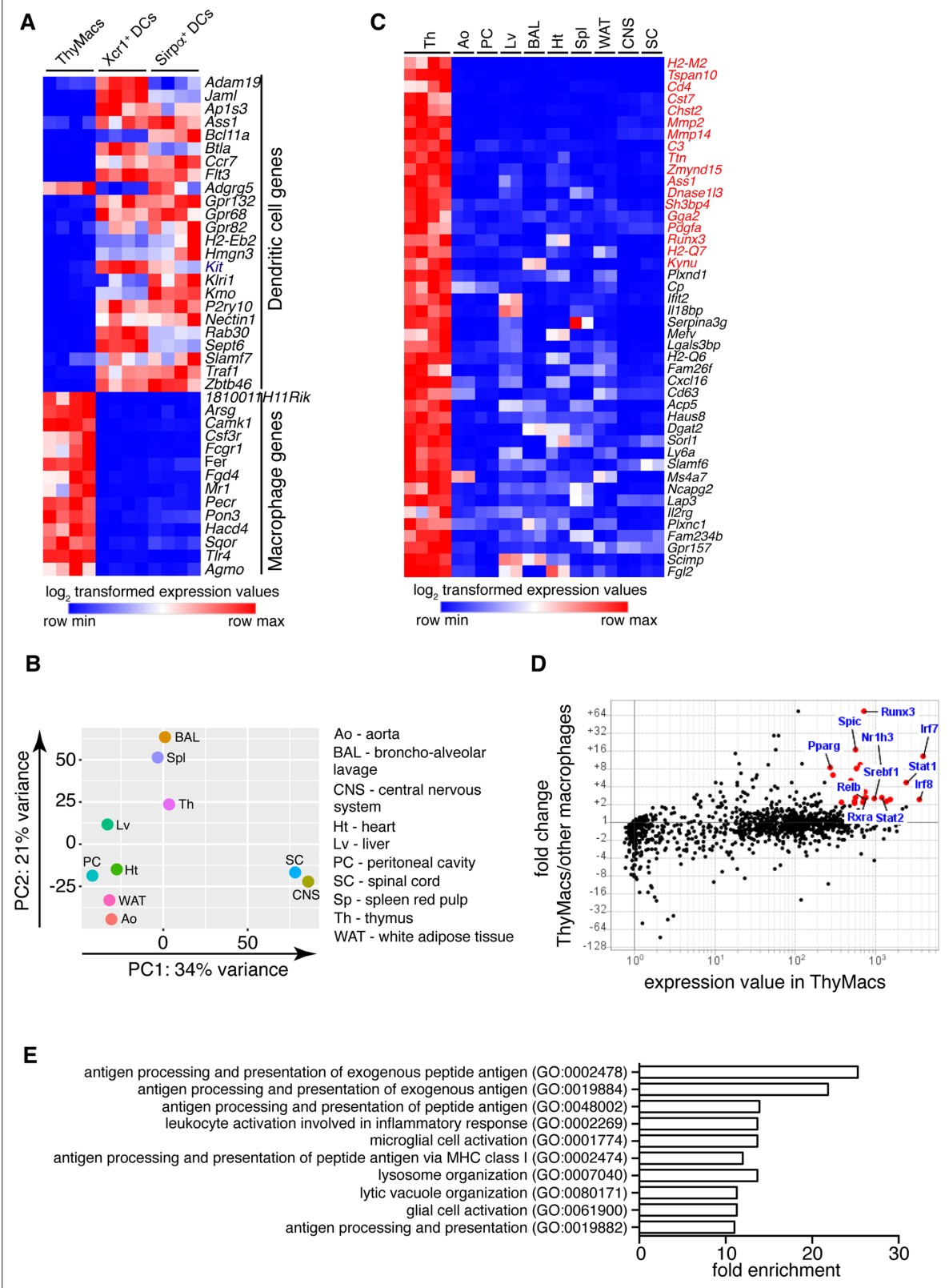

**Figure 2.** Transcriptional profile of thymic macrophages (ThyMacs). (**A**) Expression of classical dendritic cell (cDC)-specific genes (top) and macrophage-specific genes (bottom) in ThyMacs and two populations of thymic cDCs (ThyDCs) – *Xcr1⁺* ThyDCs and *Sirpa⁺* ThyDCs. (**B**) Principal components analysis of ThyMacs and nine other populations of tissue-resident macrophages in duplicates. (**C**) Highly expressed (>500) genes enriched (>fivefold) in ThyMacs (four samples) compared to nine other tissue-resident macrophage populations (two samples each). The genes in red are >10-fold up-regulated

*Figure 2 continued on next page*

*Figure 2 continued*

in ThyMacs. (**D**) Comparison of the geometric mean expression of transcription factors in ThyMacs (four samples) and the nine other macrophage populations (two samples each). Transcription factors with expression >250 and fold change >2 are marked with red dots. (**E**) Top 10 gene ontology (GO) pathways in ThyMacs based on the 500 most highly expressed genes in these cells.

The online version of this article includes the following source data and figure supplement(s) for figure 2:

**Figure supplement 1.** Example of the gating strategy to identify thymic macrophages (ThyMacs) among *Spic*GFP+ cells.

**Figure supplement 1—source data 1.** *SpicGFP* is a faithful reporter of thymic macrophages.

**Figure supplement 2.** Representative flow cytometry plots of the expression of four reporter alleles in thymic macrophages (ThyMacs; left), frequencies of GFP/YFP+ cells among ThyMacs (middle), and frequencies of ThyMacs among GFP/YFP+ cells (right).

**Figure supplement 2—source data 1.** The utility of various reporter mouse strains to identify thymic macrophages.

**Figure supplement 3.** Representative flow cytometry plots of the expression of *Spic*GFP in thymic macrophages (ThyMacs).

**Figure supplement 3—source data 1.** Expression of *SpicGFP* in thymic macrophages.

## Thymic macrophages can present antigens to T cells and clear apoptotic cells

Next, we investigated the biological functions of thymic macrophages. Our findings that these cells express MHC2 and many other genes involved in antigen presentations prompted us to test if they can efficiently activate T cells. We pulsed sorted thymic macrophages with chicken ovalbumin (Ova) and cultured them with naïve OT2 cells labeled with CFSE. The positive control, thymic DCs, efficiently induced OT2 cell proliferation, while peritoneal macrophages were very inefficient (*Figure 3A and B*), similar to other tissue-resident macrophages (*Baratin et al., 2017*). Surprisingly, thymic macrophages induced proliferation in a considerable proportion (~30%) of OT2 cells as calculated by FlowJo's Proliferation Modeling module. Thus, thymic macrophages are able antigen-presenting cells, although not as good as DCs.

To confirm the ability of thymic macrophages to clear apoptotic cells, we did in vitro engulfment assay. Thymocytes were induced to undergo apoptosis by dexamethasone treatment and labeled with pHrodo Red dye. pHrodo Red is weakly fluorescent at neutral pH, but its fluorescence increases significantly at low pH, for example, in lysosomes. Thus, engulfed apoptotic cells can be clearly identified by their strong red fluorescence. We incubated the pHrodo Red-labeled apoptotic thymocytes for 2 hr with sorted thymic or peritoneal cavity macrophages and detected the extent of efferocytosis by fluorescent microscopy. Thymic macrophages were avid phagocytes, and we could record many instances of efferocytosis at this time point (*Figure 3C and D*). However, peritoneal macrophages were able to phagocytose even more apoptotic cells.

To determine if thymic macrophages are the major phagocytes in the thymus in vivo, we evaluated their participation in the phagocytosis of apoptotic cells in the thymus by TUNEL staining. Most TUNEL+ cells could be found clearly inside or closely associated with MerTK+ and TIM4+ cells in the thymus (*Figure 3E and F*). On average, ~85% of all TUNEL+ cells were within 5 μm of MerTK+ cells, confirming that thymic macrophages are the dominant phagocytic population in the thymus (*Figure 3G*). The degree of co-localization between TUNEL+ cells and TIM4+ cells was slightly lower, ~75% on average, possibly reflecting the absence of TIM4 expression on a small proportion of thymic macrophages (*Figure 1E*).

## Expression of *Timd4* and *Cx3cr1* can distinguish two populations of thymic macrophages

Our phenotypic characterization showed clear signs of heterogeneity within thymic macrophages, including the presence of TIM4+ and TIM4− cells (*Figure 1E*) and *Cx3cr1*GFP+ and *Cx3cr1*GFP− cells (*Figure 2—figure supplement 1*). To determine the degree of thymic macrophage heterogeneity, we performed scRNA-Seq of sorted *Csf1r*GFP+ and *Cd11c*YFP+ thymic cells. *Csf1r* is required for the survival of most macrophages and is considered their definitive marker (*Witmer-Pack et al., 1993*; *Sasmono et al., 2003*), while *Cd11c*YFP is expressed in many myeloid cells, including macrophages (*Hume, 2011*). Both reporters identified an overlapping set of cells (*Figure 4—figure supplement 1*). At least seven clusters could be identified and assigned to different cell types by specific marker expression (*Figure 4A and B*), including macrophages, B cells, pDCs, contaminating thymocytes, and

**Table 1.** Expression of differentially up-regulated transcription factors in thymic macrophages. Transcription factors that were highly expressed in thymic macrophages (>250) and up-regulated >twofold in thymic macrophages compared to non-thymic macrophages were listed alphabetically, and the geometric means of four replicates of thymic macrophages (ThyMacs) and two replicates of each of the nine non-thymic macrophage populations (non-ThyMacs) were recorded. Non-thymic macrophages are: spleen red pulp macrophages, Kupffer cells, broncho-alveolar lavage macrophages, peritoneal cavity macrophages, aorta macrophages, heart macrophages, white adipose tissue macrophages, central nervous system microglia, and spinal cord macrophages.

| Gene name | ThyMacs | non-ThyMacs |
|---|---|---|
| Irf7 | 3879.32 | 300.82 |
| Irf8 | 3528.27 | 1474.35 |
| Stat1 | 2403.69 | 522.04 |
| Dnmt3a | 1515.94 | 647.81 |
| Znxf1 | 1379.89 | 635.36 |
| Stat2 | 1210.35 | 472.53 |
| Nr1h3 | 1182.17 | 147.05 |
| Srebf1 | 975.09 | 399.06 |
| Rxra | 760.26 | 298.55 |
| Trps1 | 746.36 | 232.48 |
| Runx3 | 723.14 | 9.76 |
| Relb | 715.53 | 293.92 |
| Sp100 | 696.94 | 324.47 |
| Zbp1 | 639.19 | 69.83 |
| Tfec | 588.72 | 74.66 |
| Spic | 573.11 | 34.36 |
| Nfkbie | 569.74 | 226.76 |
| Ncoa4 | 550.69 | 249.15 |
| Rest | 548.22 | 269.22 |
| Meis3 | 530.8 | 120.91 |
| Bhlhe40 | 490.59 | 99.56 |
| Parp12 | 414.11 | 126.82 |
| Arid5b | 374.03 | 177.08 |
| Creb5 | 295.14 | 47.91 |
| Pparg | 276.54 | 33.24 |

multiple cDC clusters. Two clusters expressed the macrophage/monocytes-specific TF *Mafb* and high levels of *Fcgr1* (CD64), *Mertk*, and *Adgre1* (F4/80), indicating that they are macrophages (*Figure 4— figure supplement 2*). An additional cluster expressed *Mafb* together with *Fcgr1* and *Adgre1* but not *Mertk*, fitting the description of monocytes. There was no expression of *Mafb* outside these three clusters confirming that our flow cytometry gating had identified all macrophages in the thymus. Once we zoomed onto *Mafb*-expressing cells, we could distinguish three separate populations: (1) mono-cytes that expressed high levels of *Ly6c2* and *Itgam* (CD11b) but did not express *Mertk*; (2) *Timd4*+ (encoding TIM4) macrophages that also expressed high levels of *Spic* and *Slc40a1*, but low levels of *Cx3cr1*; (3) *Cx3cr1*+ macrophages that expressed low levels of *Timd4*, *Spic*, and *Slc40a1* (*Figure 4C*

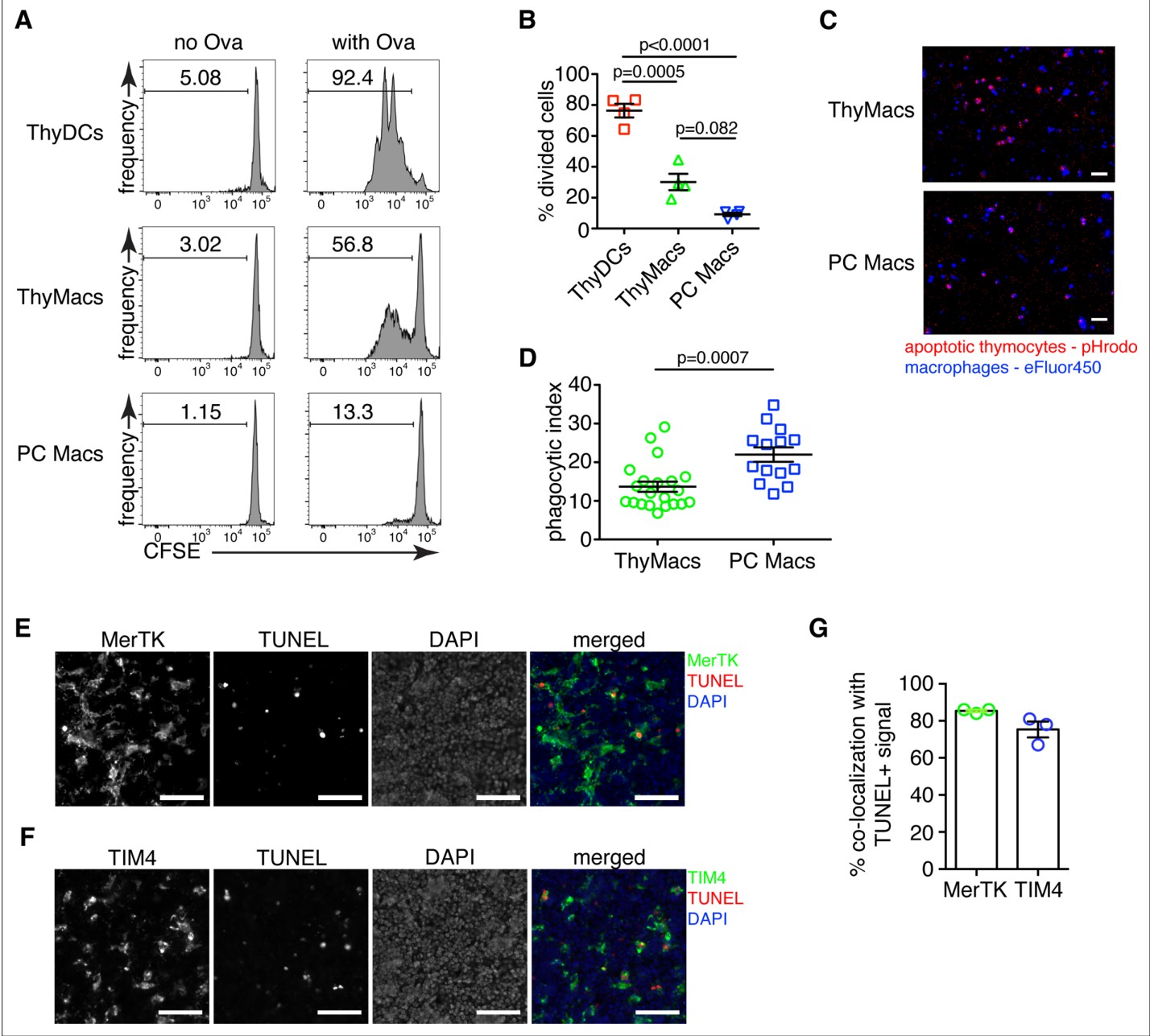

**Figure 3.** Thymic macrophages can present antigens to T cells and phagocytose apoptotic cells. (**A**) Naïve OT2 T cells were labeled with CFSE and cultured with purified thymic dendritic cells (ThyDCs), thymic macrophages (ThyMacs), or peritoneal cavity macrophages (PC Macs) in the presence or absence of chicken ovalbumin (Ova). Three days later, the CFSE dilution was assessed by flow cytometry. (**B**) Quantification of the cell division in naïve OT2 cells by using the Cell Proliferation module in FlowJo that calculates the percent of cells from the initial population that has undergone division. (**C**) Example immunofluorescent images of ThyMacs or PC Macs phagocytosis apoptotic thymocytes. The macrophages were labeled with eFluor 450, while the apoptotic thymocytes with pHrodo Red. An intense red signal within the macrophages indicates phagocytosed thymocytes. (**D**) Quantification of the percentage of macrophages that have engulfed at least one thymocyte (phagocytic index). (**E**) Example images showing co-localization of TUNEL+ apoptotic cells and MerTK+ ThyMacs in thymic sections. (**F**) Example images showing co-localization of TUNEL+ apoptotic cells and TIM4+ ThyMacs in thymic sections. Scale bars in (**E and F**) are 50 µm. (**G**) Frequencies of the co-localization of TUNEL+ signal with MerTK+ and TIM4+ cells. Flow cytometry plots in (**A**) are representative of two independent experiments. All immunofluorescent images are representative of at least three independent repeats. Data in (**B, D, and G**) represent mean ± SEM. Each symbol in **B** and **G** is an individual mouse. Each symbol in **D** is a field of view.

The online version of this article includes the following source data for figure 3:

**Source data 1.** Antigen presentation and phagocytosis of apoptotic cells by thymic macrophages.

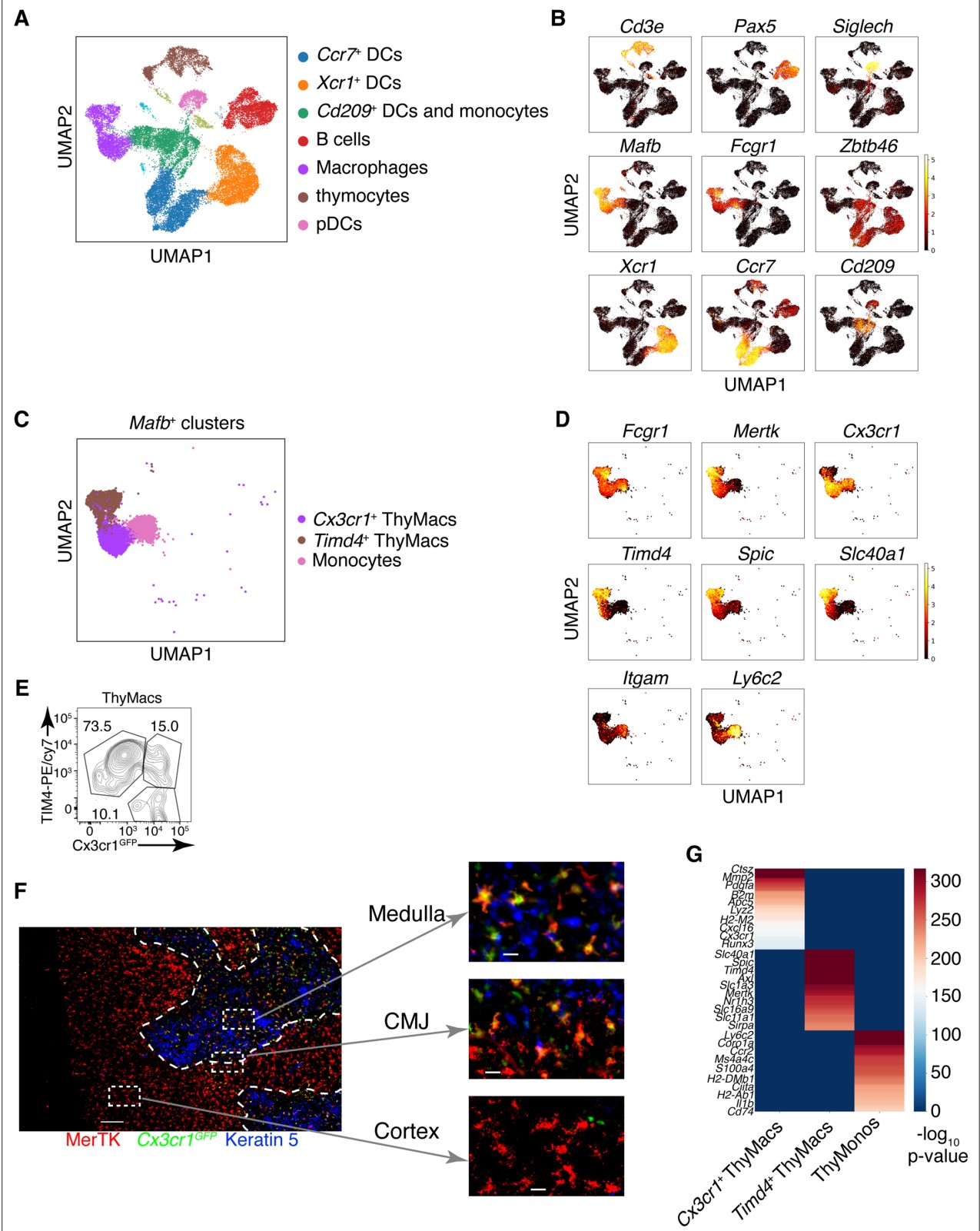

**Figure 4.** Two populations of macrophages with distinct localization exist in the thymus. (**A**) Identification of the clusters from the single-cell RNA-sequencing data based on lineage-specific markers. (**B**) Expression of lineage-specific markers in different clusters. (**C**) UMAP clusters from **A** with high expression of the transcription factor *Mafb* fall into three groups: monocytes, *Timd4*+ macrophages, and *Cx3cr1*+ macrophages. (**D**) Expression of the indicated genes in the three *Mafb*-positive clusters. (**E**) A flow cytometry plot of *Cx3cr1*GFP and TIM4 expression in thymic macrophages (ThyMacs). The

*Figure 4 continued on next page*

*Figure 4 continued*

plot is representative of >10 individual experiments. The numbers inside the plot are the percentages of the cell populations in the respective gates. (**F**) Immunofluorescent staining of the thymus of *Cx3cr1^GFP* mouse stained with MerTK (a marker for all macrophages) and Keratin 5 (a marker for medulla). The scale bar is 150 µm. Areas in the cortex, medulla, and the cortico-medullary junction (CMJ) represented by the dashed boxes are enlarged to the right to show the co-localization of *Cx3cr11^GFP* and MerTK signal in CMJ and medulla, but not in the cortex. The scale bars in the images to the right are 20 µm. The images are representative of three individual mice. (**G**) Differentially expressed genes among *Timd4^+* thymic macrophages, *Cx3cr1^+* thymic macrophages, and thymic monocytes. The negative log₁₀ p-values for the genes expressed in each cluster were calculated as described in the Materials and methods, and the top 50 differentially expressed genes were plotted in the figure. Ten of these genes are listed on the left.

The online version of this article includes the following figure supplement(s) for figure 4:

**Figure supplement 1.** UMAP clustering of the single-cell RNA-sequencing data shows that the cells from the three samples (one from GFP⁺ cells in MAFIA mice and two from YFP⁺ cells in *Cd11c^YFP* mice) overlap considerably.

**Figure supplement 2.** The relative expression of prototypical macrophage genes *Mafb*, *Fcgr1* (CD64), *Mertk*, and *Adgre1* (F4/80) among thymic cells sorted as *Csf1r^GFP+* and *Cd11c^YFP+*.

**Figure supplement 3.** Representative immunofluorescent image of WT thymic frozen section stained for TIM4, CD64, and UEA-1 (a marker for medulla).

**Figure supplement 4.** The relative expression of *Tgfb1* among thymic cells from the single-cell RNA-sequencing data.

*and D*). Both macrophages and monocytes expressed *Fcgr1* (CD64). Thus, these data indicate that thymic macrophages consist of two populations with distinct expression profiles.

We confirmed the results from scRNA-Seq by flow cytometry. We could identify discrete TIM4⁺*Cx3cr1^GFP−* and TIM4⁻*Cx3cr1^GFP+* macrophages (*Figure 4E*). There was even a TIM4⁺*Cx3cr1^GFP+* intermediate population that could not be distinguished in the scRNA-Seq dataset, likely because of the lack of statistical power. To determine the localization of the two distinct macrophage populations, we stained thymic sections from *Cx3cr1^GFP* mice with an antibody to MerTK. The *Cx3cr1^GFP−* MerTK⁺ cells correspond to *Timd4⁺* macrophages, while the *Cx3cr1^GFP+*MerTK⁺ cells would be the *Cx3cr1^GFP+* macrophages. Strikingly, the two macrophage populations showed distinct localization in young mice. *Timd4⁺* macrophages were located in the cortex, while the *Cx3cr1^GFP+* macrophages resided in the medulla and the cortico-medullary junction (*Figure 4F*). The result was confirmed with direct staining for TIM4 that showed intense signal in the cortex, particularly in the deep cortex, and absence of staining in the medulla (*Figure 4—figure supplement 3*). However, the medulla still featured many CD64⁺ macrophages.

To better understand the differences between the two populations of thymic macrophages, we looked for differentially expressed genes. We included the thymic monocytes in the comparison, as these cells clustered the closest to macrophages. *Timd4⁺* macrophages expressed the highest levels of the TFs *Spic*, *Maf*, and *Nr1h3*; the receptors for apoptotic cells *Axl*, *Mertk*, and *Timd4*; and many Slc transporters such as *Slc40a1*, *Slc1a3*, *Slco2b1*, *Slc11a1*, and *Slc7a7* (*Figure 4G* and *Table 2*). *Cx3cr1⁺* macrophages expressed high levels of the TF *Runx3*; a distinct set of phosphatidylserine receptors such as *Stab1*, *Anxa5*, and *Anxa3*; many degradative enzymes such as *Mmp2*, *Mmp14*, *Dnase1l3*, *Acp5*, *Lyz2*, *Ctsz*, *Ctss*, *Ctsd*, and *Ctsl*; cytokines such as *Pdgfa*, *Cxcl16*, and *Ccl12*; and molecules involved in MHC1 antigen presentation such as *B2m*, *H2-M2*, *H2-K1*, and *H2-Q7*. Thymic monocytes were characterized by differential expression of the typical monocyte genes *Ly6c2*, *Ccr2*, and *S100a4*, and genes involved in MHC2 antigen presentation such as *Ciita*, *H2-DMb1*, *H2-Ab1*, and *Cd74*.

## Yolk-sac progenitors contribute to embryonic thymic macrophages

The ontogeny of thymic macrophages has been examined by only one study since the realization that many tissue-resident macrophages are descendants of embryonic progenitors (*Tacke et al., 2015*). Based on *Flt3^Cre* fate-mapping, the authors concluded that most adult thymic macrophages derive from HSCs. To determine if YS progenitors contribute to embryonic thymic macrophages, we used *Cx3cr1^CreER* fate mapping (*Yona et al., 2013*). Injection of 4-OHT at E9.5 in *ROSA26^LSL-GFP* mouse mated with a *Cx3cr1^CreER* male permanently tags YS progenitors and their descendants with GFP (*Figure 5A*). Indeed, E19.5 microglia that are exclusively derived from YS progenitors were labeled to a high degree (*Figure 5B*). After adjusting for incomplete labeling based on the microglia, we found that at E15.5 >50% of thymic macrophages were fate mapped, i.e., from YS origin (*Figure 5C*). However, GFP⁺ thymic macrophages decreased to just ~11% at E19.5, suggesting that YS progenitors establish the embryonic thymic macrophage pool but are quickly replaced by subsequent wave(s) of macrophages.

**Table 2.** List of the differentially expressed genes among *Timd4*⁺ thymic macrophages, *Cx3cr1*⁺ thymic macrophages, and thymic monocytes.

The top 100 differentially expressed genes among the three clusters are listed by their negative log$_{10}$ transformed p-value.

| *Cx3cr1* + ThyMacs | | *Timd4* + ThyMacs | | ThyMonos | |
|---|---|---|---|---|---|
| Gene name | Adjusted p-value | Gene name | Adjusted p-value | Gene name | Adjusted p-value |
| Ctsz | 0 | Hpgd | 0 | Alox5ap | 0 |
| Cd63 | 0 | Serpinb6a | 0 | S100a6 | 0 |
| Pmepa1 | 0 | Slc40a1 | 0 | Ly6c2 | 0 |
| Zmynd15 | 0 | Cd81 | 0 | Ifi27l2a | 0 |
| Olfml3 | 0 | Vcam1 | 0 | Fau | 0 |
| Mmp2 | 0 | Cfp | 0 | Coro1a | 0 |
| AU020206 | 1.60E-290 | Spic | 0 | Ccr2 | 0 |
| Plxnd1 | 1.59E-285 | Trf | 0 | Rps27 | 0 |
| Cst7 | 8.68E-279 | Actn1 | 0 | Tmsb10 | 0 |
| Dnase1l3 | 2.45E-270 | Maf | 0 | Ifitm2 | 7.21E-302 |
| Timp2 | 2.15E-267 | Pld3 | 0 | Fxyd5 | 6.36E-299 |
| Lgals3bp | 8.69E-263 | Il18 | 0 | Rps19 | 2.04E-292 |
| Pdgfa | 6.87E-255 | Mrc1 | 0 | Rpl18 | 6.50E-291 |
| Mmp14 | 2.33E-253 | Crip2 | 0 | Rpl9 | 1.11E-289 |
| Fam46c | 9.99E-235 | Tmem65 | 0 | Rps23 | 1.28E-289 |
| Chst2 | 1.19E-226 | Igf1 | 0 | Napsa | 8.91E-279 |
| Cp | 5.36E-225 | Epb41l3 | 0 | Ms4a4c | 8.25E-272 |
| Camk1 | 7.12E-225 | Timd4 | 0 | Plac8 | 2.10E-270 |
| B2m | 1.09E-222 | Blvrb | 0 | Rpl18a | 9.26E-269 |
| Lhfpl2 | 4.52E-217 | Clec1b | 0 | S100a4 | 4.98E-268 |
| Acp5 | 5.90E-216 | Cd68 | 0 | Cd52 | 3.67E-267 |
| Lag3 | 3.91E-213 | Axl | 0 | Rps14 | 1.94E-266 |
| Lyz2 | 1.28E-209 | Paqr9 | 3.32E-307 | Ifitm3 | 3.19E-263 |
| H2-M2 | 1.22E-199 | Sdc3 | 3.45E-305 | Rpl34 | 2.02E-261 |
| Psap | 7.26E-198 | Myo9a | 5.59E-305 | Rps27a | 3.67E-260 |
| Gatm | 1.33E-192 | Scp2 | 3.79E-302 | Rpl36 | 1.54E-259 |
| Cpd | 1.50E-192 | Selenop | 2.10E-295 | Rps16 | 2.55E-258 |
| C3 | 2.34E-187 | Lrp1 | 2.08E-294 | Rpl24 | 1.37E-257 |
| Cxcl16 | 8.11E-183 | Lap3 | 1.45E-290 | Rps9 | 6.34E-253 |
| Lgals3 | 1.57E-182 | Marcks | 2.77E-279 | Gpr141 | 1.21E-246 |
| Ube2j1 | 1.63E-180 | Glul | 3.64E-279 | Rpl27a | 3.06E-243 |
| Plxnc1 | 9.84E-180 | Hebp1 | 3.76E-278 | Rpl17 | 8.15E-241 |
| Stab1 | 4.07E-176 | Ear2 | 4.53E-276 | Rps24 | 1.46E-240 |
| Cyth1 | 3.27E-163 | Apoc1 | 2.49E-275 | Rps13 | 2.34E-236 |
| Spsb1 | 3.96E-163 | Kcna2 | 3.72E-275 | Rpl38 | 1.95E-226 |

*Table 2 continued on next page*

*Table 2 continued*

| *Cx3cr1* + ThyMacs | | *Timd4* + ThyMacs | | ThyMonos | |
|---|---|---|---|---|---|
| Blnk | 2.35E-162 | Myo10 | 9.05E-269 | H2-DMb1 | 1.02E-223 |
| Cx3cr1 | 9.29E-162 | Atp13a2 | 2.95E-267 | Rps18 | 5.39E-223 |
| Med10 | 5.25E-161 | Slc1a3 | 6.24E-263 | Rpl19 | 3.68E-221 |
| Nek6 | 5.28E-160 | Slco2b1 | 1.11E-258 | Rpl8 | 2.01E-219 |
| Ptms | 1.05E-159 | mt-Nd2 | 3.45E-258 | Rpl7a | 4.17E-217 |
| Anxa5 | 1.10E-156 | Wwp1 | 2.16E-253 | Gm34084 | 5.23E-216 |
| Gpnmb | 1.21E-154 | Aplp2 | 4.22E-248 | Rpl13 | 2.08E-215 |
| Itgb5 | 2.78E-154 | Atp8a1 | 5.03E-248 | Rpl11 | 2.47E-213 |
| Myo5a | 1.11E-146 | P2ry13 | 3.17E-247 | Rpl35a | 2.13E-210 |
| Runx3 | 1.81E-146 | Ccdc148 | 4.70E-245 | Rpsa | 1.62E-209 |
| Tmem176a | 2.34E-144 | Grn | 1.58E-244 | Rpl6 | 5.70E-208 |
| Ctss | 4.81E-141 | Bank1 | 1.82E-239 | Tpt1 | 2.63E-206 |
| Sh3pxd2b | 9.38E-141 | Mertk | 2.15E-238 | Rack1 | 2.14E-203 |
| Rtcb | 4.42E-140 | Nr1h3 | 1.13E-235 | Rpl23 | 6.14E-199 |
| Fam20c | 1.91E-139 | Prnp | 2.93E-235 | Rpl26 | 7.48E-198 |
| Il2rg | 8.84E-138 | Ninj1 | 2.42E-234 | Rps6 | 6.64E-197 |
| Lpcat2 | 8.53E-137 | Fcna | 3.33E-233 | Rps10 | 2.06E-195 |
| Kynu | 8.49E-136 | Csrp1 | 1.16E-230 | Ier5 | 1.06E-191 |
| Tnfsf13b | 8.77E-136 | Rgl1 | 7.18E-229 | Rps3 | 8.23E-185 |
| Gpr157 | 1.18E-135 | Lpl | 4.94E-223 | Rpl27 | 8.23E-185 |
| Tgfbr1 | 7.63E-135 | Fam213b | 1.08E-222 | Rps5 | 8.36E-185 |
| H2-K1 | 1.15E-133 | Tcf7l2 | 1.26E-222 | Rps7 | 3.96E-182 |
| Basp1 | 1.23E-133 | AB124611 | 4.64E-221 | Rps15a | 6.82E-182 |
| Pla2g7 | 1.80E-132 | Abcc3 | 3.28E-216 | Rps11 | 1.97E-180 |
| Fth1 | 4.19E-131 | Fcgrt | 5.79E-216 | Rps4x | 5.07E-180 |
| Ggh | 1.85E-126 | Tgm2 | 1.88E-215 | Rplp0 | 3.09E-177 |
| Adam19 | 6.94E-126 | Itgad | 5.35E-214 | Ly6i | 8.17E-176 |
| C3ar1 | 7.35E-125 | Ptgs1 | 2.94E-213 | S100a11 | 6.23E-175 |
| Ccl12 | 3.37E-123 | Laptm4a | 1.01E-212 | Atox1 | 1.22E-174 |
| Hvcn1 | 2.51E-121 | Comt | 1.33E-206 | Pim1 | 9.56E-174 |
| Anxa3 | 8.60E-121 | Creg1 | 3.24E-205 | Sh3bgrl3 | 3.97E-173 |
| Tgfbi | 1.88E-120 | Adgre1 | 9.67E-205 | Ciita | 7.35E-173 |
| Ctsd | 2.73E-117 | Clec12a | 6.33E-204 | Eef1a1 | 6.09E-172 |
| Itm2c | 5.19E-116 | Tspan4 | 7.80E-203 | Rps3a1 | 9.09E-168 |
| Tmem119 | 5.62E-116 | Txn1 | 9.13E-203 | Gm2a | 6.07E-165 |
| Rap2a | 1.03E-114 | Ctsb | 9.52E-201 | Ptprc | 2.05E-163 |
| Ctsl | 4.00E-114 | Mrap | 5.65E-197 | Rpl37 | 1.51E-161 |
| Itga6 | 1.83E-113 | Slc16a9 | 5.99E-197 | Rps25 | 3.03E-160 |
| B4galnt1 | 2.45E-113 | Abcg3 | 3.83E-196 | H3f3a | 5.92E-159 |

*Table 2 continued on next page*

*Table 2 continued*

| *Cx3cr1* + ThyMacs | | *Timd4* + ThyMacs | | ThyMonos | |
|---|---|---|---|---|---|
| Fam3c | 1.64E-112 | Pla2g15 | 4.22E-196 | Btg2 | 1.14E-158 |
| Tmem173 | 1.54E-111 | C1qc | 6.17E-192 | Rpl15 | 1.42E-158 |
| Ski | 3.59E-111 | Agpat3 | 1.68E-191 | Cnn2 | 1.09E-156 |
| Anpep | 5.85E-111 | Hs6st1 | 1.95E-191 | Cdkn1a | 2.57E-156 |
| Gng2 | 2.37E-110 | Dmpk | 2.15E-191 | Slfn1 | 4.83E-155 |
| Nceh1 | 2.88E-110 | Cd38 | 1.79E-190 | Sem1 | 4.08E-154 |
| H2-Q7 | 4.94E-108 | Tmem26 | 2.02E-189 | Lsp1 | 1.34E-152 |
| Rtn1 | 1.28E-106 | Slc11a1 | 1.05E-188 | Rpl37a | 1.78E-152 |
| Sorl1 | 1.31E-103 | Cd300a | 1.41E-187 | Rpl22 | 3.64E-152 |
| Glipr1 | 1.22E-102 | Slc7a7 | 3.28E-187 | Sirpb1c | 4.81E-152 |
| Gsn | 2.00E-102 | Cyb5a | 6.94E-187 | Traf1 | 6.97E-152 |
| Afdn | 4.54E-102 | Sipa1l1 | 7.41E-187 | Emb | 4.22E-151 |
| Ak2 | 1.11E-101 | Il18bp | 1.48E-186 | Rpl30 | 1.32E-147 |
| Ntpcr | 2.21E-98 | Cd86 | 2.52E-183 | Rps15 | 1.14E-146 |
| Scarb2 | 3.16E-97 | Vamp5 | 3.05E-183 | H2-Ab1 | 2.84E-145 |
| Creb5 | 5.41E-97 | Jup | 6.69E-182 | Il1b | 3.05E-145 |
| Gsto1 | 5.56E-97 | Blvra | 1.30E-178 | Rps28 | 4.52E-145 |
| Ncf1 | 4.26E-96 | Mgst1 | 6.48E-178 | Jarid2 | 1.82E-143 |
| Ppfia4 | 4.97E-96 | Tbxas1 | 1.47E-177 | Rps26 | 1.53E-142 |
| Chchd10 | 7.77E-96 | Hpgds | 2.04E-177 | Rpl32 | 4.21E-142 |
| Gna12 | 1.23E-95 | Tgfbr2 | 2.70E-176 | Pld4 | 9.07E-142 |
| Mvb12b | 1.80E-95 | Clec4n | 3.52E-175 | Cbfa2t3 | 1.54E-141 |
| Rasal3 | 1.45E-94 | Ms4a7 | 5.30E-175 | Rps21 | 4.04E-141 |
| Scoc | 6.86E-94 | Sirpa | 3.35E-171 | Fgr | 4.04E-141 |
| Cfb | 6.00E-93 | Fyn | 2.84E-168 | Rps8 | 1.11E-139 |
| Lmna | 1.04E-92 | Cadm1 | 2.20E-167 | Cd74 | 5.34E-138 |

## Differential contribution of adult bone marrow-derived monocytes to the two thymic macrophage populations

To investigate the possibility that thymic macrophages arise from adult bone marrow-derived monocytes, we devised two complementary experiments. First, we evaluated the contribution of circulating adult monocytes to thymic macrophages without the confounding effect of radiation damage on the thymus. We created shield chimeras by subjecting CD45.2 mice to a lethal dose of irradiation while protecting their upper body and the thymus with a 5-cm lead shield, followed by reconstitution with CD45.1 bone marrow (**Figure 5D**). We analyzed *Timd4*+ and *Cx3cr1*+ thymic macrophages separately after 6 weeks because we suspected they might have different origins. As CX3CR1 protein expression was low on thymic macrophages (**Figure 5B**), we defined the *Cx3cr1*+*Timd4*− population as TIM4−. The donor-derived monocytes in the blood were, on average, 57%, but less than 2% of TIM4+ thymic macrophages were CD45.1+ (**Figure 5E and F**), suggesting very limited contribution of adult circulating monocytes to the TIM4+ macrophage pool. The percentage of HSC-derived TIM4− macrophages (on average 23%) was intermediate between the monocytes and TIM4+ macrophages, pointing out that a sizeable part of TIM4− cells was derived from adult HSCs.

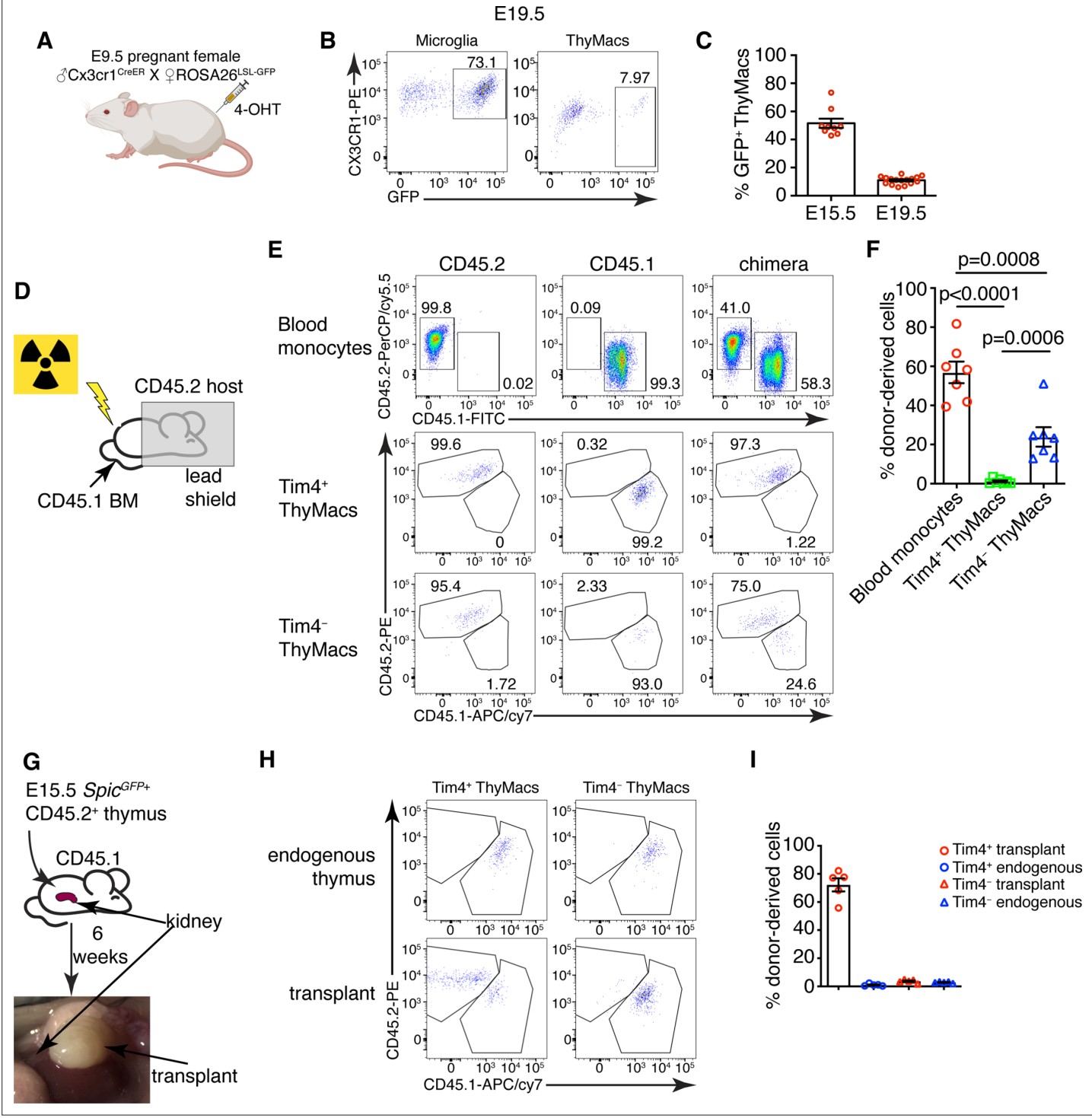

**Figure 5.** Yolk sac (YS), non-YS-derived embryonic progenitors, and adult hematopoietic stem cells sequentially contribute to the thymic macrophage (ThyMac) pool. (**A**) Scheme of the YS-progenitor labeling experiments. E9.5 pregnant *ROSA26*[LSL-GFP] mice mated with *Cx3cr1*[CreER] males were injected with 4-hydroxytamoxifen (4-OHT) and sacrificed at E15.5 or E19.5. (**B**) Representative flow cytometry plots of the *Cx3cr1*[GFP] expression in microglia (CD45[+]CD11b[+] cells in the brain) and ThyMacs of the pups at E19.5. (**C**) Frequencies of GFP[+] ThyMacs at E15.5 and E19.5 adjusted to the degree of labeling of microglia. (**D**) Scheme of the shield chimera experiments. Congenic CD45.2 mice were lethally irradiated with their upper body protected by a 5-cm thick lead shield and then injected with CD45.1[+] bone marrow. (**E**) Representative flow cytometric analysis of CD115[+]CD11b[+] blood monocytes, TIM4[+] and TIM4[−] ThyMacs for donor-derived (CD45.1[+]) and host-derived (CD45.2[+]) cells. Non-chimeric CD45.1 and CD45.2 samples serve as controls for the gating. (**F**) Frequencies of donor-derived blood monocytes, TIM4[+] and TIM4[−] ThyMacs. (**G**) Scheme of the thymus transplantation experiments.

*Figure 5 continued on next page*

*Figure 5 continued*

Embryonic thymuses from E15.5 *Spic^GFP+* CD45.2^+ mice were transplanted under the kidney capsule of CD45.1^+ mice and analyzed 6 weeks later. (**H**) Representative flow cytometry plots of donor- (CD45.2^+) vs. host (CD45.1^+)-derived TIM4^+ and TIM4^− ThyMacs in the transplanted thymus. The host thymus (endogenous thymus) serves as a negative control. (**I**) Frequencies of CD45.2^+ (donor-derived) cells among TIM4^+ and TIM4^− ThyMacs in the transplanted and endogenous thymuses of the mice. Data in **C**, **F**, and **I** are mean ± SEM with two litters, seven, and five mice per group, respectively. The numbers in the flow cytometry plots are the percent of cells in the respective gates. Each symbol in the graphs is an individual mouse or embryo.

The online version of this article includes the following source data and figure supplement(s) for figure 5:

**Source data 1.** Contribution of different waves of progenitors to the thymic macrophage pool.

**Figure supplement 1.** Representative flow cytometry plots of donor- (CD45.2^+) vs. host (CD45.1^+)-derived thymocytes in the transplanted thymus.

**Figure supplement 1—source data 1.** All thymocytes in the transplanted thymus are derived from host hematopoietic stem cells.

**Figure supplement 2.** Comparison of the geometric mean fluorescent intensities (gMFI) of *Spic^GFP* TIM4^+ and TIM4^− thymic macrophages of host and donor-origin in the transplanted thymus.

**Figure supplement 2—source data 1.** Only TIM4^+ thymic macrophages in the transplanted thymus express *Spic^GFP* and are donor-derived.

We also transplanted E15.5 thymuses from *Spic^GFP+* CD45.2 embryos under the kidney capsule of adult CD45.1 mice and analyzed them 6 weeks later (*Figure 5G*). By that time, >99% of thymocytes in the transplanted thymus were derived from CD45.1^+ host HSCs, indicating successful replacement by HSC-derived progenitors (*Figure 5—figure supplement 1*). TIM4^− thymic macrophages were derived entirely from host HSCs, just like thymocytes. In contrast, most TIM4^+ cells (on average 70%) were donor-derived (*Figure 5H and I*). Moreover, only CD45.2^+ TIM4^+ macrophages expressed *Spic^GFP* (*Figure 5—figure supplement 2*). As expected, thymic macrophages in the endogenous thymus were all CD45.1^+. The results from our transplantation experiments show that the progenitors of most TIM4^+ thymic macrophages are of embryonic origin, while TIM4^− cells are derived from adult monocytes. Altogether our results suggest that the two populations of thymic macrophages have different origins. TIM4^+ cells are derived from embryonic precursors and can survive long term without much contribution from adult HSC and monocytes. In contrast, TIM4^− thymic macrophages rely mostly on adult HSCs for their generation and replacement.

## Thymic macrophages can proliferate in situ

TIM4^+ macrophages can persist for many weeks in the thymus without constant replacement from blood monocytes, suggesting they can divide in situ. Staining for the proliferation marker Ki67 revealed that ~4% of all thymic macrophages expressed this marker compared to an isotype control (*Figure 6A and B*). To prove that thymic macrophages are proliferative, we tested the incorporation of the nucleotide analog 5-ethynyl-2′-deoxyuridine (EdU). Short-term EdU labeling experiments unexpectedly revealed that thymic macrophages become EdU^+ with faster kinetics than thymocytes (*Figure 6—figure supplement 1*). The most likely explanation for this puzzling result is that some of the thymic macrophages have engulfed apoptotic thymocytes that have recently divided and incorporated EdU. Thus, EdU could have accumulated in these macrophages through phagocytosis, not cell division. To circumvent this caveat, we designed a pulse-chase experiment (*Figure 6C*). Mice were injected daily with EdU for 21 days so that all cells that proliferated in that period would incorporate the label. Most thymocytes and thymic macrophages became EdU^+ at d. 21 (*Figure 6D*). After 21 more days of 'chase period', only ~0.2% of thymocytes had retained the EdU label, consistent with the existence of a tiny population of long-term resident thymocytes consisting mainly of regulatory T cells and NKT cells (*McCaughtry et al., 2007*; *Figure 6D and E*). However, ~5% of the thymic macrophages were EdU^+, suggesting they divided during the labeling period. We also sorted thymic macrophages and subjected them to cell cycle analysis. Although almost all thymic macrophages were in G0/G1 phase, a small population of ~3% was in the G2/M phase of the cell cycle (*Figure 6F and G*). Surprisingly, most *Mki67*^+ thymic macrophages belonged to the *Cx3cr1*^+ subset, and only a few of the *Timd4*^+ cells were positive (*Figure 6H*). We confirmed this result from the scRNA-Seq analysis experimentally. The expression of Ki-67 was significantly higher in TIM4^− than in TIM4^+ thymic macrophages (*Figure 6G*), suggesting that the former is the more proliferative subset. Collectively, four independent approaches documented that a small proportion (3–5%) of thymic macrophages are actively dividing under homeostatic conditions within the thymus. The majority of the dividing cells were from the adult HSC-derived *Cx3cr1*^+ subset. *Timd4*^+ macrophages were primarily quiescent.

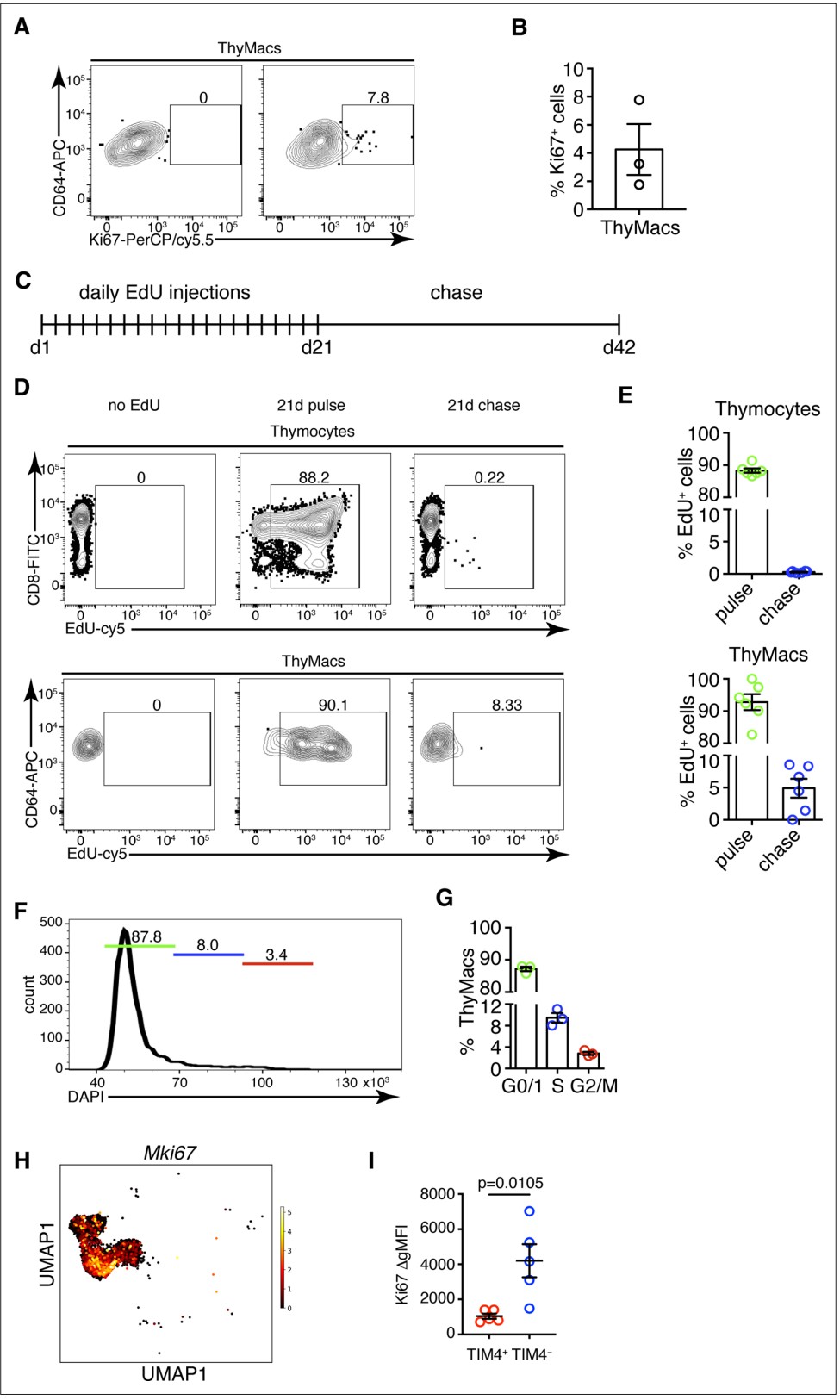

**Figure 6.** Thymic macrophages (ThyMacs) exhibit a low degree of proliferation. (**A**) Example flow cytometry plots of Ki67 staining of ThyMacs. (**B**) Frequency of Ki67+ ThyMacs. (**C**) Scheme of 5-ethynyl-2'-deoxyuridine (EdU) pulse/chase experiment: mice were injected daily with 1 mg EdU i.p. for 21 days and rested for 21 more days. (**D**) Example flow cytometry plots of EdU staining of thymocytes (upper row) and ThyMacs (lower row). (**E**) Frequencies

*Figure 6 continued on next page*

*Figure 6 continued*

of EdU$^+$ cells among thymocytes (top graph) and ThyMacs (bottom graph). (**F**) Example flow cytometry plot of cell cycle analysis of FACS-sorted ThyMacs. (**G**) Frequencies of ThyMacs in different stages of the cell cycle. (**H**) UMAP plot of *Mki67* expression in *Mafb*-positive clusters from the single-cell RNA-sequencing data described in *Figure 4*. (**I**) Comparison of Ki67 protein expression in TIM4$^+$ and TIM4$^-$ ThyMacs. The expression is measured as the difference of the geometric mean fluorescent intensities of the Ki67 antibody staining and isotype control (ΔgMFI). The numbers in the flow cytometry plots are the percent of cells in the respective gates. Data are mean ± SEM from three mice (**B and G**) or five mice (**I**), or six to seven individual mice (**E**). Each dot is an individual mouse. Statistical significance in (**I**) was determined by unpaired Student's t-test.

The online version of this article includes the following source data and figure supplement(s) for figure 6:

**Source data 1.** Low degree of proliferation in thymic macrophages mainly restricted to the TIM4- subset.

**Figure supplement 1.** Example flow cytometry plots of the 5-ethynyl-2'-deoxyuridine (EdU) accumulation in thymocytes and thymic macrophages 2 hr after 1 mg EdU i.p. or vehicle injection.

---

## *Cx3cr1*$^+$ cells give rise to *Timd4*$^+$ cells during embryonic development

To determine if the two populations of thymic macrophages are related, we first analyzed the kinetics of their appearance during embryonic development. At the earliest time point (E14.5), all thymic macrophages were *Cx3cr1*$^+$, and only ~20% were also TIM4$^+$ (*Figure 7A and B*). The proportion of TIM4$^+$ cells increased at E17.5, and TIM4$^+$*Cx3cr1*$^-$ cells started to appear. In the neonatal period, almost all macrophages were TIM4$^+$, and very few remained TIM4$^-$. The proportion of TIM4$^-$ cells increased in 6 weeks old mice, but TIM4$^+$ macrophages remained the dominant population. These kinetics (*Figure 7C*) are consistent with *Timd4*$^+$ macrophages developing from *Cx3cr1*$^+$ cells before birth. Another plausible scenario is that distinct progenitors give rise to different thymic macrophage populations (e.g. YS progenitors give rise to *Cx3cr1*$^+$*Timd4*$^-$, and HSC-derived progenitors develop into *Timd4*$^+$ macrophages). To test the latter hypothesis, we revisited the fate mapping of YS progenitors (*Figure 5A*). Although a larger part (~60% at E15.5) of fate-mapped cells was *Cx3cr1*$^+$TIM4$^-$ cells (*Figure 7D*), a substantial proportion (~40% at E15.5) of fate-mapped TIM4$^+$ macrophages could clearly be identified at both E15.5 and E19.5, suggesting that YS progenitors can give rise to both *Cx3cr1*$^+$ and *Timd4*$^+$ cells. Thus, the simplest explanation for our findings is that *Timd4*$^+$ cells develop from *Cx3cr1*$^+$ cells during embryonic development. This transition is complete in the first week after birth as there were essentially no *Cx3cr1*$^+$TIM4$^-$ thymic macrophages remaining at d.7 (*Figure 7A and B*). To formally demonstrate that *Cx3cr1*$^+$ macrophages can give rise to *Timd4*$^+$ cells during embryonic development, we injected 4-OHT in E15.5 pregnant females carrying *Cx3cr1*$^{CreER}$ × *ROSA26*$^{LSL-GFP}$ fetuses (*Figure 7E*). At this time almost all thymic macrophages are *Cx3cr1*$^+$ (*Figure 7A*). Just before birth, at E19.5, we could identify a sizeable population of TIM4$^+$CX3CR1$^-$ among fate-mapped cells, suggesting that they originate from *Cx3cr1*$^+$ progenitors (*Figure 7F and G*).

## *Cx3cr1*$^+$ thymic macrophages slowly accumulate with age at the expense of *Timd4*$^+$ cells

To understand the dynamics of the two resident thymic macrophage populations with age, we induced recombination in *Cx3cr1*$^{CreER}$ × *ROSA26*$^{LSL-GFP}$ mice during the neonatal period (*Figure 8A*) or at 6 weeks of age (*Figure 8C*) and compared the proportion of GFP$^+$ cells 3 and 42 days after labeling. The extent of labeling of TIM4$^+$ thymic macrophages did not change within these 6 weeks, no matter whether the mice were treated with tamoxifen in the first week after birth or at 6 weeks (*Figure 8B and D*), suggesting an absence of a significant influx from unlabeled cells (e.g. monocytes). In contrast, the proportion of labeled TIM4$^-$ thymic macrophages decreased significantly 6 weeks after tamoxifen injection in neonatal and adult mice, suggesting that this population was diluted by unlabeled cells. To further substantiate these findings, we examined older WT mice and found out that the proportions of TIM4$^-$ thymic macrophages increased with age, and in mice >8 months old, they accounted for ~70% of all macrophages in the organ (*Figure 8E*). As these changes in the proportions of the thymic macrophage subpopulations occurred at the background of thymic involution, we wanted to know if the accumulation of TIM4$^-$ cells was only relative or also in absolute cell numbers. In contrast to TIM4$^+$ thymic macrophages that reached peak numbers at an early age and then declined significantly, TIM4$^-$ cells tended to increase their numbers in older mice (*Figure 8F*). Thus, we conclude

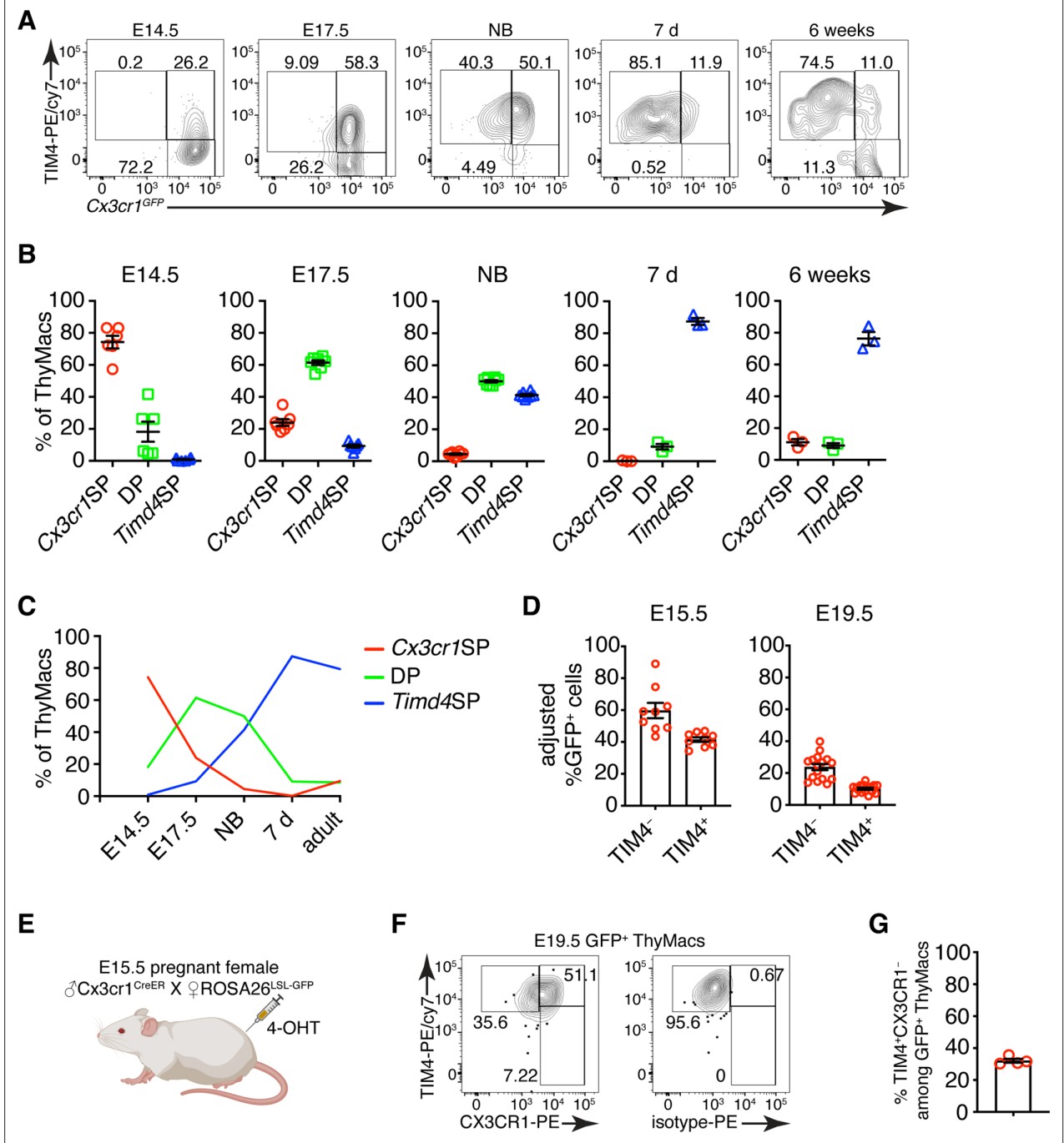

**Figure 7.** *Timd4*+ thymic macrophages are derived from *Cx3cr1*+ cells during embryonic development. (**A**) Example flow cytometry plots for the expression of *Cx3cr1GFP* and TIM4 on thymic macrophages at different times during embryonic development (E14.5, E17.5), immediately after birth, at 7 days, and 6 weeks of age. (**B**) Frequencies of *Timd4*+*Cx3cr1*− (*Timd4* single-positive or *Timd4*SP), *Timd4*+*Cx3cr1*+ (double-positive or DP), and *Cx3cr1*+*Timd4*− (*Cx3cr1* single-positive or *Cx3cr1*SP) thymic macrophages at the indicated time points. (**C**) Kinetics of the changes in different subpopulations of thymic macrophages from E14.5–6 weeks. (**D**) Frequencies at E15.5 and E19.5 of GFP-labeled cells among TIM4+ or TIM4− cells in *Cx3cr1CreER* × *ROSA26LSL-GFP* embryos treated with 4-OHT at E9.5. (**E**) Scheme of the fate-mapping experiments showing the relationship between *Cx3cr1*+ and *Timd4*+ thymic macrophages during embryonic development. E15.5 pregnant *ROSA26LSL-GFP* mice mated with *Cx3cr1CreER* males were injected with 4-hydroxytamoxifen (4-OHT) and sacrificed at E19.5. (**F**) Representative flow cytometry staining for TIM4 and CX3CR1 in fate-mapped GFP+ thymic macrophages at E19.5. The panel to the right is the isotype control for CX3CR1-PE staining. (**G**) Frequencies of TIM4+CX3CR1− cells among fate-mapped GFP+ macrophages. Data are shown as meanSEM and are from at least two independent experiments for each panel. Each symbol is an individual mouse or embryo.

*Figure 7 continued on next page*

*Figure 7 continued*

The online version of this article includes the following source data for figure 7:

**Source data 1.** Thymic macrophages during the embryonic period.

that, after birth, the numbers of TIM4[+] macrophages follow the kinetics of the thymus size – increase in young and decrease in old mice, and they are not replaced by other cells. In contrast, since the first week of life, *Cx3cr1*[+] cells are recruited to the thymus, accumulate with age, and in old mice, form the predominant phagocytic population in the organ.

## Discussion

Here, we have described the phenotype, transcriptional profile, localization, diversity, ontogeny, and maintenance of macrophages in the thymus. These cells express the typical macrophage markers CD64, MerTK, and F4/80 and are transcriptionally most similar to splenic red pulp macrophages and liver Kupffer cells. However, they have a unique expression profile dominated by genes involved in antigen presentation and lysosomal degradation. We found that thymic macrophages consist of two populations with distinct localization. *Timd4*[+] macrophages occupied the cortex, while *Cx3cr1*[+] cells were located in the medulla and the cortico-medullary junction. While YS-derived macrophages dominated the early stages of thymus development, they were quickly replaced by non-YS embryonic progenitors that gave rise to the *Timd4*[+] thymic macrophages that persisted into adulthood and formed the main macrophage population in young mice. *Cx3cr1*[+] macrophages slowly accumulated after birth and became the most abundant population in old mice.

Altogether our data depict thymic macrophages as typical tissue-resident macrophages originating from multiple hematopoietic waves, surviving long term, and expressing the core macrophage-specific genes. They are most similar transcriptionally to splenic red pulp macrophages and Kupffer cells, which is not surprising considering that they all specialize in efferocytosis and have efficient lysosomal degradation machinery. These three populations also shared expression of the TF *Spic* that is induced by heme released following red blood cells phagocytosis (*Haldar et al., 2014*). However, the thymus is not known as a place for erythrocyte degradation. Thus, the mechanism for *Spic* up-regulation in thymic macrophages is unclear.

The unique features of thymic macrophages include high expression of genes involved in the IFN-I pathway, antigen presentation, and lysosomal degradation. The up-regulation of IFN-I-stimulated genes such as *Stat1*, *Stat2*, *Irf7*, and *Irf8* can be explained by the constitutive secretion of IFN-I by thymic epithelial cells (*Lienenklaus et al., 2009*; *Otero et al., 2013*). The purpose of IFN-I expression in the thymus in the absence of a viral infection is unclear. Still, one possibility is that it mediates negative selection to IFN-dependent genes as part of central tolerance.

Thymic macrophages highly express molecules involved in antigen presentation, including MHC1 and MHC2, although the latter is expressed at lower levels than in cDCs, and are functionally competent to induce T cell activation. Thus, they have the potential to present antigens for both negative selection and agonist selection. These two activities have traditionally been assigned solely to cDCs (*Breed et al., 2018*). However, recent evidence suggests that negative selection is most efficient when the cell that presents the antigen to an auto-reactive thymocyte is also the one that phagocytoses it (*Kurd et al., 2019*). So, macrophages' participation in thymocyte selection needs to be re-evaluated.

The extraordinary ability of thymic macrophages to engulf and degrade apoptotic thymocytes has been appreciated for a long time (*Surh and Sprent, 1994*), and our RNA-Seq data provides additional supporting evidence for this function by highlighting the up-regulation of pathways involved in lysosomal degradation. Moreover, we recently showed that the pentose phosphate pathway has a central role in buffering the efferocytosis-associated oxidative stress in thymic macrophages (*Tsai et al., 2022*). An interesting topic for future research would be understanding how the metabolites derived from apoptotic cells are returned to the microenvironment to support the proliferation of immature thymocytes. A *SoLute Carrier* (*Slc*) genes-based program has been described in vitro (*Morioka et al., 2018*), but its relevance to tissue-resident macrophages remains to be determined. Altogether, our study demonstrates that thymic macrophages are a unique subset of tissue-resident macrophages and support the idea that resident macrophage phenotype is determined by the combination of ontogeny, microenvironment, and other factors (*Blériot et al., 2020*).

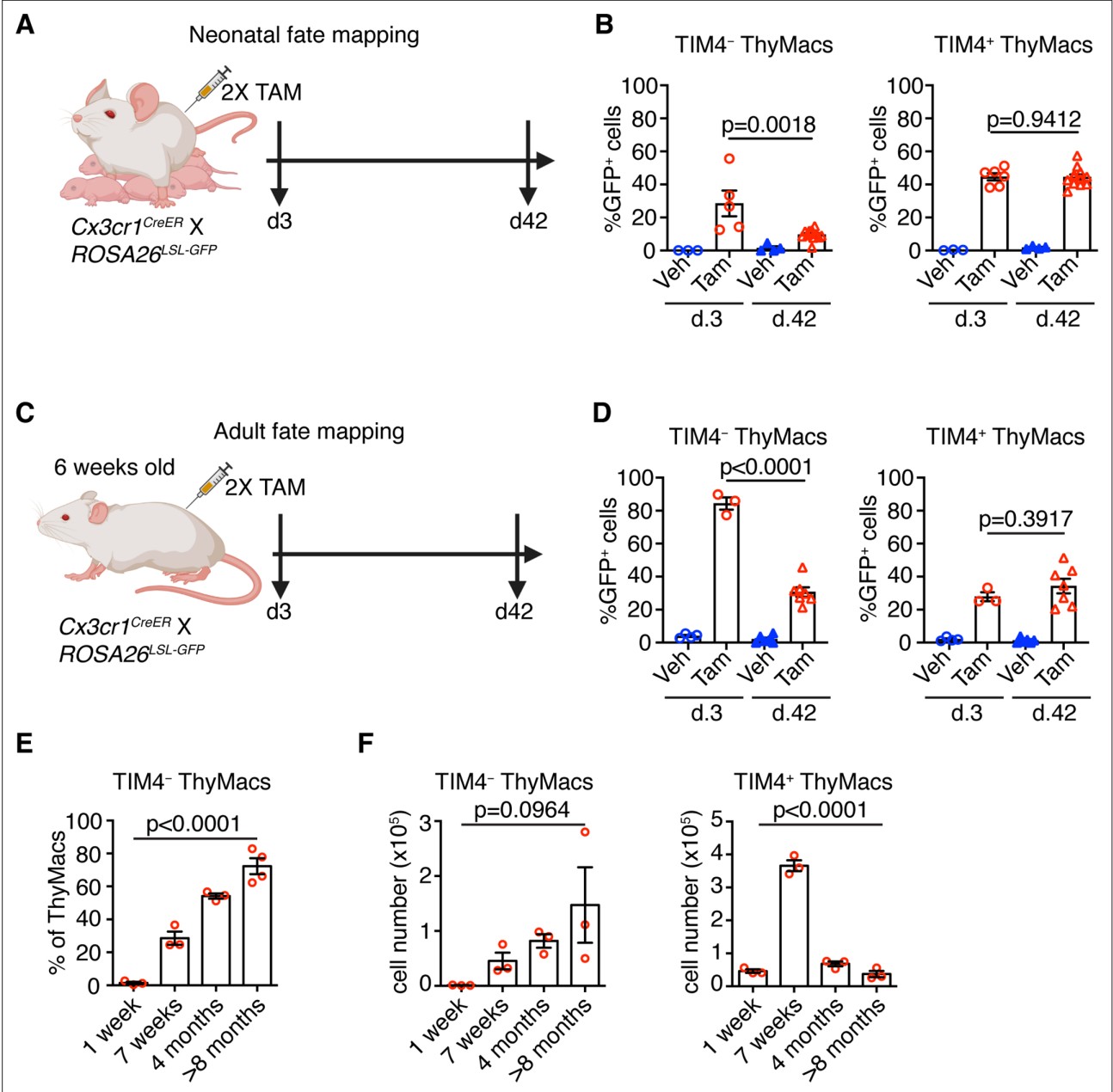

**Figure 8.** *Timd4*+ thymic macrophages (ThyMacs) are progressively lost, while *Cx3cr1*+ cells slowly accumulate with age. (**A**) Scheme of the neonatal fate mapping: A nursing dam was injected twice with tamoxifen (Tam) or vehicle (Veh) in the first week after giving birth to *Cx3cr1*CreER × *ROSA26*LSL-GFP pups. The mice were sacrificed 3 or 42 days after the last injection, and the degree of labeling of TIM4+ and TIM4– ThyMacs was examined by flow cytometry. (**B**) Frequencies of GFP+ TIM4+ or TIM4– ThyMacs from neonatally fate-mapped mice after 3 and 42 days. Vehicle-injected nursing dam litters (Veh) served as a control for non-specific labeling. (**C**) Scheme of the adult fate mapping: Six weeks old *Cx3cr1*CreER × *ROSA26*LSL-GFP mice were injected twice with Tam or Veh. The mice were sacrificed 3 or 42 days after the last injection, and the degree of labeling of TIM4+ and TIM4– ThyMacs was examined by flow cytometry. (**D**) Frequencies of GFP+ TIM4+ or TIM4– ThyMacs from adult fate-mapped mice after 3 and 42 days. (**E**) Frequencies of TIM4– ThyMacs at different ages. (**F**) Changes in the numbers of TIM4– and TIM4+ ThyMacs with age. The data are mean ± SEM from two independent experiments (**B**) or at least three individual mice per time point (**D**, **E**, and **F**). Each symbol is an individual mouse. Statistical significance in the difference between Tam-treated samples at different time points was determined with unpaired Student's t-test (**B and D**). One-way ANOVA was used to assess the significance of the change in TIM4+ and TIM4– ThyMacs percentages and numbers with age (**E and F**).

The online version of this article includes the following source data for figure 8:

**Source data 1.** Dynamics of TIM4+ and TIM4- thymic macrophages with age.

Together with the study by *Tacke et al., 2015*, our work builds the following model for thymic macrophage origin: thymic macrophages develop in three distinct waves: YS-derived progenitors dominate the early stages of thymus development but are replaced before birth by a second wave of YS-independent embryonic progenitors that forms the bulk of thymic macrophages after birth and can self-maintain into adulthood. With age, there is a slow and steady influx of $Timd4^-Cx3cr1^+$ macrophage precursors that occupy the medulla and cortico-medullary junction, becoming the major phagocytic population in the thymus of older mice (>8 months). The second wave of YS-independent macrophages is most likely the progeny of embryonic HSCs based on $Flt3^{Cre}$ fate mapping that showed that >80% of thymic macrophages in adult mice are descendants of HSCs (*Tacke et al., 2015*), whether HSC-independent fetal liver monocytes contribute to thymic macrophages and to what extent awaits the creation of models that can specifically target this population of progenitors. Recruitment of circulating monocytes to the resident macrophage pool in the thymus has been ruled out previously by parabiosis and $Ccr2^{-/-}$ mice (*Tacke et al., 2015*). Our shield chimera experiments have arrived at similar conclusions. However, the relatively short duration of these experiments and their focus on the bulk thymic macrophages has prevented the recognition of the gradual accumulation of $Timd4^-$ macrophages. Once we zoomed in on this minor cell population in young mice, the fate mapping clearly showed an influx of unlabeled progenitors. Whether the progenitors of $Timd4^-$ macrophages are monocytes remains to be formally demonstrated. However, monocytes have been singled out as the source of all macrophage populations exhibiting replacement in adults examined to date (*Molawi et al., 2014*; *Goldmann et al., 2016*; *Jacome-Galarza et al., 2019*; *Tamoutounour et al., 2013*; *Bain et al., 2014*; *Bain et al., 2016*). An alternative possibility involves thymocyte progenitors that, under certain circumstances, have been shown to differentiate into macrophages and granulocytes in the thymus (*Wada et al., 2008*; *Bell and Bhandoola, 2008*). However, if this occurs in unmanipulated mice at a steady state remains unclear.

We observed interesting dynamics of the $Cx3cr1^+$ macrophages in the thymus. Thymic macrophage progenitors are initially $Cx3cr1^+$ during the embryonic period but gradually down-regulate this chemokine receptor and up-regulate $Timd4$ so that by day 7 after birth, there are almost no $Cx3cr1^+Timd4^-$ cells remaining. $Cx3cr1^+Timd4^-$ macrophages start to increase after the neonatal stage, but these cells come from an entirely different source – adult hematopoietic cell-derived progenitors – and slowly accumulate in the medulla with time so that by 6–8 months, they are the majority of the resident macrophages in that tissue. Both YS-derived primitive macrophages and fetal liver monocytes express $Cx3cr1$ (*Hoeffel et al., 2012*; *Mass et al., 2016*). However, the tissue-resident macrophages in some organs (e.g. Kupffer cells, alveolar macrophages, red pulp macrophages, and Langerhans cells) lose $Cx3cr1$ expression similar to thymic macrophages, while the macrophages in the intestines, aorta, kidney, dermis, lymph node T cell zone, and microglia do not (*Yona et al., 2013*; *Tamoutounour et al., 2013*; *Ensan et al., 2016*; *Baratin et al., 2017*). Similar processes may occur in other tissues where the embryonic macrophages transition to a $Cx3cr1^-$ phenotype and are slowly replaced by monocyte-derived cells with age. However, detailed time-course analyses of $Cx3cr1$ expression starting before birth and extending to very old (1 year) mice coupled with lineage tracing would be necessary to document if this transition takes place.

The spatial segregation of the two macrophage populations in the thymus implies that they might have distinct functions. $Timd4^+$ cells are restricted to the cortex and are particularly abundant in the deeper cortex, close to the medulla. Both positive and negative selection of thymocytes occur there, so we speculate that $Timd4^+$ macrophages might be specialized in efferocytosis of $CD4^+CD8^+$ (double-positive) thymocytes that cannot interact with cortical thymic epithelial cells and die by neglect or are auto-reactive and undergo clonal deletion in the cortex (*Stritesky et al., 2013*). On the other hand, $Cx3cr1^+$ macrophages accumulate in the medulla – the thymic region specialized in negative selection to tissue-restricted antigens (TRAs). They might contribute to the process in several ways: (1) by carrying TRAs from blood and peripheral organs. A similar process has been described for cDC2 (SIRPα$^+$ DCs) (*Bonasio et al., 2006*; *Baba et al., 2009*). In fact, $Cx3cr1^+$ thymic macrophages could have contributed to this role because they were not distinguished from cDC2 in this study. (2) By capturing TRAs from $Aire^+$ medullary thymic epithelial cells and presenting them to auto-reactive thymocytes as shown for DCs (*Gallegos and Bevan, 2004*; *Koble and Kyewski, 2009*; *Voboril et al., 2020*). (3) By phagocytosing apoptotic TRA-specific medullary thymocytes, a process we have

observed before (*Kurd et al., 2019*). The exact involvement of thymic macrophages in the selection events in the thymus remains to be determined.

The accumulation of the *Cx3cr1*+ cells in older mice has clear implications for thymus aging. One key feature of thymus involution is the accumulation of extracellular matrix produced by fibroblasts and the emergence of white adipocytes (*Dixit, 2012*). A well-recognized driver of fibrosis is TGFβ1 (*Budi et al., 2021*) that is induced by efferocytosis in macrophages (*Huynh et al., 2002*). *Tgfb1* was highly expressed in thymic macrophages. However, its expression was the highest in the *Timd4*+ subset (*Figure 4—figure supplement 4*). This expression pattern casts some doubt that this molecule is the primary driver of extracellular matrix accumulation during thymic involution because *Timd4*+ macrophages peak in young mice (*Figure 8F*). At that time, there is minimal extracellular matrix in the cortex where these cells reside. In addition, during thymic involution, the number of these cells declines significantly. The clear correlation between the accumulation of *Cx3cr1*+ thymic macrophages and thymic involution suggests that some factor(s) produced exclusively by these cells would be more relevant. For example, *Cx3cr1*+ thymic macrophages are the predominant producer of the growth factor PDGFα (*Figure 4G*) that is required for the maintenance of adipocyte stem cells and can stimulate tissue fibrosis (*Rivera-Gonzalez et al., 2016*; *Olson and Soriano, 2009*). The gradual accumulation of *Cx3cr1*+ macrophages could increase the availability of PDGFα in the aging thymus stimulating extracellular matrix production and differentiation of precursors into adipocytes. This model predicts that limiting the influx of *Cx3cr1*+ macrophage precursors could delay thymus involution.

Recent work described a novel phagocytic and antigen-presenting cell type in the thymus called monocyte-derived DCs (*Vobořil et al., 2020*). The phenotype of these cells overlaps with the CD64+F4/80loCD11b+ cells in our study. However, we favor the classification of these cells as monocytes based on their expression of *Mafb*, CD64, and Ly6C and lack of expression of the defining DC TF *Zbtb46* (*Figure 4B and D*; *Satpathy et al., 2012*). As monocytes can differentiate into cDC2, particularly in the context of inflammation (*Guilliams et al., 2018*), the precise identity and relationship of this population to thymic cDC2 remain to be established.

In the past several years, scRNA-Seq has come to the forefront of biologists' efforts to disentangle the cellular diversity of tissues. Several comprehensive studies have included samples from mouse or human thymus (*Han et al., 2018*; *Tabula et al., 2018*; *Tabula, 2020*). However, too few thymic macrophages were sampled in these studies to give meaningful clustering results. Efforts specifically targeting the thymus have provided considerably more information (*Kernfeld et al., 2018*; *Park et al., 2020*), but macrophage diversity was still not recognized. Characterization of rare populations such as thymic macrophages (~0.1% of all cells in the thymus) requires optimized enzymatic digestion procedures and enrichment strategies, as has already been demonstrated for thymic epithelial cells (*Bornstein et al., 2018*; *Bautista et al., 2021*). Our scRNA-Seq dataset provides a rich resource for the unbiased characterization of myeloid cells in the thymus and will greatly aid in the understanding of the myeloid landscape of the thymus.

In summary, our work comprehensively characterizes macrophages in the thymus and paves the way for the exploration of their functions.

## Materials and methods
### Mice

C57BL/6Narl (CD45.2) mice were purchased from the National Laboratory Animal Center, Taipei, Taiwan (NLAC stock# RMRC11005). MAFIA (MAcrophage Fas-Induced Apoptosis, Jackson Labs stock# 005070) (*Burnett et al., 2004*), *Cx3cr1*GFP (Jackson Labs stock# 005582) (*Jung et al., 2000*), *Spic*GFP (Jackson Labs stock# 025673) (*Haldar et al., 2014*), *Cx3cr1*CreER (Jackson Labs stock# 020940) (*Yona et al., 2013*), and *B6.SJL-Ptprca Pepcb/BoyJ* (CD45.1, Jackson Labs stock# 002014) mice were purchased from the Jackson Laboratories. *Cd11c*YFP (Jackson Labs stock# 008829) (*Lindquist et al., 2004*) and *Lyz2*GFP (*Faust et al., 2000*) mice have been described. Mice ubiquitously expressing GFP from the *ROSA26* locus were generated by breeding *Pdgfra*Cre (Jackson Labs stock# 013148) (*Roesch et al., 2008*) and *ROSA26*LSL-ZsGreen (also known as *ROSA26*LSL-GFP or Ai6, Jackson Labs stock# 007906) mice (*Madisen et al., 2010*) (both from the Jackson Laboratories). A mouse from this cross was identified, in which the STOP cassette was deleted in the germline. It was designated *ROSA26*GFP and subsequently bred to C57BL/6 mice. All mice were used at 4–10 weeks of age unless otherwise specified.

Mice were bred and maintained under specific pathogen-free conditions at the animal facility of National Yang Ming Chiao Tung University (NYCU). All experimental procedures were approved by the Institutional Animal Care and Use Committee (IACUC) of NYCU.

## Treatment with 5-ethynyl-2'-deoxyuridine

Mice were i.p. injected with 1 mg EdU (Carbosynth) dissolved in PBS daily for 21 days and then rested for 21 more days. Cells from the thymus were harvested on day 21 or 42. In some experiments, the mice were sacrificed 2 hr after the first EdU injection.

## Shield chimera generation

WT (CD45.2) mice were anesthetized by i.p. injection of 120 µg/g body weight Ketamine hydrochloride (Toronto Research Chemicals) and 12 µg/g body weight Xylazine hydrochloride (Sigma). Anesthetized mice were taped to a 5-cm thick lead block so that the lead block covered the head and the chest down to the bottom of the rib cage. Then, they were irradiated with a lethal dose (1000 rad) from a $^{137}$Cs source (Minishot II, AXR) so that only their abdomen and hind legs were exposed. After recovery from anesthesia, the mice were transfused i.v. with $10^7$ bone marrow cells from a congenic (CD45.1) donor. Then, they were given trimerin (0.5 mg/mL sulfadiazine + 0.1 mg/mL trimethoprim, China Chemical and Pharmaceutical Co., Tainan, Taiwan) in the drinking water for the first 2 weeks after the irradiation and analyzed after 6 weeks.

## Cell isolation from thymus, blood, and peritoneal cavity

Thymocytes were obtained by mechanical disruption of the thymus with a syringe plunger. For myeloid cell isolation, mouse thymuses were cut into small pieces and digested with 0.2 mg/mL DNase I (Roche) and 0.2 mg/mL collagenase P (Roche) in complete DMEM for 20 min at 37°C with frequent agitation. In some experiments, thymic myeloid cells were enriched by 57% Percoll PLUS (GE Healthcare) discontinuous gradient centrifugation at 4°C, 1800 rpm, for 20 min without brake. Cells at the interface were collected and washed with PBS to remove residual silica particles. Then the cells were resuspended in PBS with 0.5% BSA (HM Biological), filtered through a 70 µm filter, and kept on ice.

Blood was isolated by cardiac puncture of sacrificed mice and immediately diluted with PBS. After centrifugation, the cell suspensions were treated with ammonium chloride-potassium lysis buffer for 3 min on ice once or twice. Peritoneal cavity cells were obtained by lavage with 5 mL PBS + 2 mM EDTA (Merck). Following gentle massage, the cavity was opened with an abdominal incision, and lavage fluid was collected.

## Flow cytometry

Single-cell suspensions (0.5–2×10$^6$ cells) from thymus, blood, or peritoneal cavity were blocked with supernatant from 2.4G2 hybridoma (a kind gift by Dr. Fang Liao, Academia Sinica, Taipei, Taiwan) and stained with fluorochrome- or biotin-labeled antibodies for 20 min on ice in PBS + 0.5% BSA + 2 mM EDTA + 0.1% NaN$_3$ (FACS buffer). The following antibodies were used: CD11b (clone M1/70), MHC2 (M5/114.15.2), CD11c (N418), F4/80 (BM8), CD115 (AFS98), SIRPα (P84), CD45 (30-F11), NK1.1 (PK136), TIM4 (RMT4-54), Gr-1 (RB6-8C5), CD64 (X54-5/7.1), Siglec H (551), Ly6C (HK1.4), CD3ε (145–2 C11), CD8α (53–6.7), CD19 (6D5), B220 (RA3-6B2), CD4 (GK1.5), CD51 (RMV-7), CD45.1 (A20), CD45.2 (104), CX3CR1 (SA011F11), and EpCAM (G8.8) from BioLegend; Axl (MAXL8DS), MerTK (DS5MMER), and Ki67 (SolA15) were from eBioscience; Siglec F (E50-2440), CD90.2 (30-H12), and CD11c (HL3) were from BD Biosciences. Cells were washed, and if necessary, incubated for 20 more minutes with fluorochrome-labeled streptavidin: streptavidin-AF647 (Jackson Immunoresearch) or streptavidin-APC/cy7, streptavidin-BV421, streptavidin-BV605 (BioLegend). After the last wash, the cells were resuspended in FACS buffer containing DAPI (BioLegend), Propidium Iodide (Sigma), or DRAQ7 (BioLegend) and analyzed immediately on an LSR Fortessa flow cytometer running Diva 8 software (BD Biosciences). Typically, 500,000 cells were collected from thymus samples. Data were analyzed using FlowJo software (TreeStar).

For intracellular staining, after surface antibody staining, the cells were labeled with Zombie Aqua (BioLegend) for 30 min in ice. Then, the cells were fixed with 2% paraformaldehyde (Electron Microscope Sciences) in PBS for 20 min on ice, permeabilized with either 0.5% Triton-X 100 (Sigma) for

20 min on ice, or with Foxp3 staining kit (eBioscience) according to the protocol provided by the manufacturer and stained with antibodies for intracellular markers for 40–60 min on ice.

For cell cycle analysis, $1–5×10^5$ sorted thymic macrophages were fixed with 70% ethanol for 2 hr on ice. The cells were spun down at 1800 rpm for 20 min at 4°C, washed with PBS, and stained with 1 µg/ml DAPI (BioLegend) for 30 min at room temperature in the dark.

For EdU staining, after surface marker and Zombie Aqua staining, cells were fixed with 2% paraformaldehyde in PBS for 20 min on ice and permeabilized with 0.5% Triton X-100 in PBS at room temperature for 20 min. EdU was detected by adding an equal volume of 2× Click reaction buffer consisting of 200 mM Tris, 200 mM ascorbic acid (Acros), 8 mM $CuSO_4$ (Acros), and 8 µM Cy5-azide (Lumiprobe) to the permeabilized cells resuspended in 0.5% Triton X-100 in PBS and incubated at room temperature for 30 min. Cells were washed twice with 0.5% Triton X-100 in PBS and analyzed on a flow cytometer.

## Cell sorting

The sorting of thymic macrophages was done following the IMMGEN guidelines. Briefly, the thymuses of three male C57BL/6Narl mice were harvested in ice-cold staining buffer containing phenol red-free DMEM (Gibco) with 10 mM HEPES (Sigma), 0.1% $NaN_3$, and 2% FBS (Gibco). Single-cell suspensions were prepared as described in the Flow cytometry section. Percoll PLUS was used to enrich mononuclear cells. The cells were resuspended at $10^8$ /mL in staining buffer and labeled with appropriate antibodies for 15 min in ice. To sort thymic macrophages, the cells were first labeled with biotinylated antibodies to lineage markers (Lin) – CD3, CD8, Gr1, and B220. After washing, the cells were stained with antibodies to CD11b, F4/80, CD45, CD64, and Streptavidin-APC/cy7 for 15 min in ice. For sorting thymus XCR1+ and SIRPα+ cDCs, antibodies to XCR1, SIRPα, CD11c, MHC2, CD64, and F4/80 were used. For sorting peritoneal cavity macrophages, antibodies to ICAM2 and F4/80 were used. Immediately before sorting, the dead cells were excluded with DRAQ7 or PI. For RNA sequencing experiments, the cells were double-sorted on FACS Melody, or Aria cell sorters (BD Biosciences), and 1000 cells were directly deposited in TCL buffer (Qiagen), frozen in dry ice, and sent to IMMGEN for RNA sequencing. Four biological replicates were prepared. For cytospin and cell cycle analysis, $1–5×10^5$ cells sorted on FACS Melody were collected in staining buffer.

## Cytospin

Sorted cells were mounted on Superfrost PLUS slides (Thermo Scientific) using a Cytospin centrifuge (Cytospin 3, Shandon) for 5 min at 500 rpm. Cells were fixed with 2% paraformaldehyde for 10 min at room temperature and stained with the Hemacolor Rapid Staining Kit (Merck Millipore). Images were collected on BX61 upright microscope (Olympus) using ×100 objective with immersion oil and captured with a CCD camera. Images were then analyzed and processed with ImageJ (NIH) and Adobe Photoshop 5.5 (Adobe).

## In vitro phagocytosis assay

$10^7$ Thymocytes were cultured in cDMEM in the presence of 1 µM of dexamethasone (Sigma) in a 3.5-cm culture dish at 37°C in 5% $CO_2$ incubator for 8 hr. Apoptosis levels were assessed by PI (Biolegend) and Annexin V-FITC (Biolegend) staining. Typically, more than 80% of cells were Annexin V+. The dexamethasone-treated thymocytes were stained with 1 µg/mL pHrodo Red, SE (ThermoFisher) in PBS for 30 min at room temperature. The cells were washed two times with cDMEM and resuspended at $2×10^6$ cells/mL. $4×10^4$ sorted peritoneal and thymic macrophages were stained with 5-µM eFluor 450 (Thermo Fisher) in PBS for 10 min at 37°C, washed two times with cDMEM, and cultured in 96-well flat-bottom culture plate (Nunc) in 100 µL cDMEM at 37°C in 5% $CO_2$ incubator. After 3 hr of attachment, the non-adherent cells were removed, and 200 µL ($4×10^5$) apoptotic thymocytes were added to the macrophages. The cells were incubated at 37°C in 5% $CO_2$ incubator for 2 hr. Fluorescent images were captured with AxioObserver 7 (Carl Zeiss) wide-field microscope equipped with Plan Apochromat 40 × NA = 1.0 objective (Zeiss) and AxioCam 702 monochrome camera (Zeiss) controlled by Zen 2.3 Blue (Zeiss) software. Image analysis was performed with Imaris 8.0.2 (Bitplane). Phagocytosis was scored by investigators blinded to the samples' identities.

## In vitro antigen presentation assay

$3\times10^4$ sorted thymic CD64⁻MHCII⁺CD11c⁺ dendritic cells, thymic, or peritoneal macrophages were cultured in 96-well round-bottom culture plate in 100 μL cDMEM at 37°C in 5% $CO_2$ incubator for 3 hr to attach. Splenocytes from OT2 mice were stained with biotinylated antibodies to CD8a, CD11b, CD11c, B220, and MHCII (all from BioLegend), washed, and labeled with anti-biotin microbeads (Miltenyi) plus CD44 microbeads (Miltenyi) in cRPMI. The cells were separated on MACS LS columns (Miltenyi) according to the manufacturer's instructions. Enriched cells (naïve CD4 T cells) were stained with 10 μM CFSE (Sigma) for 5 min in PBS at 37°C and cocultured with the sorted thymic MHCI-I⁺CD11c⁺ dendritic cells, thymic, or peritoneal macrophages, in the presence or absence of 0.5-mg/mL OVA protein (Sigma) in cRPMI at 37°C in 5% $CO_2$ incubator for 72 hr. The cells were collected and stained with antibodies to TCRβ and CD4 (from BioLegend) for flow cytometry analyses of CFSE dilution. The data were analyzed with FlowJo's Proliferation Modeling module (BD Biosciences).

## RNA sequencing analysis

RNA sequencing was done at IMMGEN using Smart-seq2 protocol (*Picelli et al., 2013*; *Picelli et al., 2014*) on a NextSeq500 sequencer (Illumina). Following sequencing, raw reads were aligned with STAR to the mouse genome assembly mm10 and assigned to specific genes using the GENCODE vM12 annotation. Gene expression was normalized by DESeq2 (*Love et al., 2014*) and visualized by Morpheus (https://software.broadinstitute.org/morpheus). The principal component analysis was done by plotPCA() function of R package 'DESeq2'. Gene expression of mouse TFs (*Schmeier et al., 2017*) was visualized in MultiplotStudio of GenePattern (*Reich et al., 2006*). GO enrichment was calculated and visualized in R by using clusterProfiler (*Yu et al., 2012*).

## Timed pregnancies and embryonic thymus analysis

To set up timed pregnancies, each male mouse (*Cx3cr1^CreER/CreER*, *Cx3cr1^GFP/GFP*, or C57BL/6) and female mouse (*ROSA26^LSL-GFP/LSL-GFP* or C57BL/6) were housed together in the same cage for one night and separated on the next day, which we defined as embryonic day 0.5 (E0.5). Female mice were assumed to be pregnant if their weight gain was over 2 g at E8.5 (*Heyne et al., 2015*). Thymuses from E14.5 and E17.5 embryos, neonatal, 1-week-old pups, and adult mice (older than 6 weeks old) were harvested, mechanically dissociated with plastic sticks in 1.5-mL centrifuge tubes, and enzymatically digested with 0.2 mg/mL DNase I and 0.2 mg/mL collagenase P in complete DMEM for 20 min at 37°C with frequent agitation. The cells were resuspended in PBS with 0.5% BSA, filtered through a 70 μm filter, kept on ice, and used flow cytometric analysis as described in the Flow cytometry section.

## Genetic fate mapping – E9.5, neonatal, and adult

For genetic fate mapping, timed pregnancies of *Cx3cr1^CreER/CreER* male and *ROSA26^LSL-GFP/LSL-GFP* female mice were set up as described. To label the *Cx3cr1⁺* erythromyeloid progenitors derived from embryonic YS (*Mass et al., 2016*), 4-hydroxytamoxifen (4-OHT from Sigma) was administered i.p. to pregnant females on E9.5 at a dose of 75 μg/g (body weight). To improve the survival of embryos and reduce the risk of abortions, progesterone (Sigma) was co-injected at a dose of 37.5 μg/g (body weight) (*Iturri et al., 2017*). To label the *Cx3cr1⁺* thymic macrophages in *Cx3cr1^CreER* × *ROSA26^LSL-GFP* neonates and adult mice, tamoxifen (TAM from Sigma) was injected i.p. at a dose of 2 mg/mouse to lactating dams on postnatal day 3 and 4 (P3 and P4) or to adult mice for two consecutive days. Thymuses were harvested and analyzed 3 days or 6 weeks after the last injection by flow cytometry.

## scRNA-Seq – sorting, library generation, and sequencing

scRNA-Seq was performed at the Genomics Center for Clinical and Biotechnological Applications of NCFB (NYCU, Taipei, Taiwan). Briefly, the thymuses of one female MAFIA and two male *Cd11c^YFP* mice were harvested and enzymatically digested as described previously. Mononuclear cells were enriched by 57% Percoll PLUS discontinuous centrifugation, washed to remove silica particles, and resuspended at $10^6$ /mL in PBS with 0.04% BSA. The cell suspensions were filtered through Falcon 35 μm strainer (Corning) and stained with viability dye (PI or DAPI) immediately before sorting. Cell sorting was performed on a FACS Melody sorter (BD Biosciences) running FACS Chorus (BD Biosciences) software in purity mode. $3\times10^5$ GFP or YFP positive cells under the live/singlet gating were collected into 5-ml round bottom tubes pre-coated with 0.04% BSA in PBS. Sorted cells were washed and resuspended

in 300 μL PBS with 0.04% BSA and then filtered again into 1.5 mL DNA LoBind tubes (Eppendorf) through a 35 μm strainer. The viability of the cells was evaluated by Countess II (Invitrogen) and Trypan Blue (ThermoFisher), and samples with cell viability rates higher than 85% were used for encapsulation and library preparation. Single-cell encapsulation and library preparation were performed using Single Cell 3' v3/v3.1 Gene Expression solution (10× Genomics). All the libraries were processed according to the manufacturer's instruction and sequenced on NovaSeq 6000 (Illumina) platform at the NHRI (Zhubei, Taiwan). Post-processing and quality control were performed by the NYCU Genome Center using the CellRanger package (v. 3.0.2, 10× Genomics). Reads were aligned to mm10 reference assembly. Primary assessment with CellRanger reported 9973 cell-barcodes with 11,385 median unique molecular identifiers (UMIs, transcripts) per cell and 3076 median genes per cell sequenced to 71.0% sequencing saturation with 94,260 mean reads per cell for MAFIA mouse sample; 9801 cell-barcodes with 13,467 median UMIs per cell and 3211 median genes per cell sequenced to 74.9% sequencing saturation with 119,820 mean reads per cell for the first *Cd11c^{YFP}* mouse sample; 12,938 cell-barcodes with 14,439 median UMIs per cell and 3199 median genes per cell sequenced to 71.4% sequencing saturation with 108,585 mean reads per cell for the second *Cd11c^{YFP}* mouse sample.

## Analysis of scRNA-Seq

### Preprocessing

The Scanpy (*Wolf et al., 2018*) pipeline was used to read the count matrix. Three batches of samples (one from GFP$^+$ cells from MAFIA mouse and two from YFP$^+$ cells from *Cd11c^{YFP}* mice) were preprocessed independently and integrated later. Cells that expressed <200 genes and genes that were expressed in <3 cells were filtered out. The percentage of mitochondrial genes was calculated, and cells with >10% mitochondrial genes were removed. Cells with >7000 genes or <1000 genes were also removed. Read counts were normalized to library size 10,000 and log-transformed with scanpy. pp.log1p function.

### Dataset integration and batch effect correction

Read count matrices and spliced/unspliced matrices were merged first. Principal component analysis was applied to reduce dimensions to 70. BBKNN (*Polański et al., 2020*) was then used to remove batch effects with the scanpy.external.pp.bbknn function with default parameters.

### Visualization and clustering

UMAP (*McInnes et al., 2018*) provided by scanpy was used to visualize data with default parameters. K-nearest neighbor and Leiden clustering were applied sequentially to cluster cells into groups. K-nearest neighbor graph construction was done by scanpy.pp.neighbors with parameters n_neighbors = 12 and n_pcs = 70. Leiden clustering was then performed by scanpy.tl.leiden with parameter resolution = 0.15. To improve UMAP visualization, scanpy.tl.paga was applied, and we trimmed unnecessary graph edges by scanpy.tl.paga with threshold = 0.018.

### Marker genes and statistics

Wilcoxon rank-sum tests were applied to examine differentially expressed genes. Clusters were selected from the result of Leiden clustering. Differentially expressed genes of a cluster against other clusters were identified by scanpy.tl.rank_genes_groups and scanpy.pl.rank_genes_groups. p-Values were collected for each cluster and transformed by negative $\log_{10}$ for better visualization. The top 50 differentially expressed genes were visualized in the figure.

## Immunofluorescent staining

Dissected thymus lobes from C57BL/6 mice were cleaned of connective tissue and fixed in 4% paraformaldehyde (Sigma) for 1 hr at 4°C, washed in PBS, submerged in 10% sucrose, and then in 30% sucrose for 12 hr each. The tissue was then frozen in Tissue-Tek OCT compound (Sakura Fintek) for cryostat sectioning. 10- or 20-μm thick sections were prepared with CryoStar NX50 (ThermoFisher) on Superfrost PLUS (ThermoScientific) microscope slides, dried overnight, and stored at –80°C until used. Before staining, the sections were fixed with acetone (Sigma) at –20°C for 10 min, air-dried, then blocked with 5% goat serum + 5% donkey serum (both from Jackson Immunoresearch) in PBS for 2 hr

and stained with primary antibodies: rat monoclonal to MerTK (DS5MMER, eBioscience), rat monoclonal to TIM4 (RMT4-54, Bio-X-Cell), rabbit polyclonal to CD64 (Sinobiological), or rabbit polyclonal to Keratin 5 (BioLegend) overnight at 4°C in a humidified chamber. After washing in PBS, the sections were labeled with goat anti-rat-Alexa Fluor 647 (Invitrogen) or goat anti-rat Cy3 (Jackson Immunoresearch) and donkey anti-rabbit Cy3 or donkey anti-rabbit AF647 (both from Jackson Immunoresearch) secondary antibodies for 2 hr at room temperature, followed by 5 min staining with DAPI. TUNEL assay was done with the Click-iT Plus TUNEL assay Alexa Fluor 647 kit (Invitrogen) according to the manufacturer's recommendations. Positive (pre-incubation with DNase I for 30 min at room temperature) and negative (no TdT enzyme) controls were always included. The sections were mounted with 0.1% n-propyl gallate (Sigma) in glycerol (Sigma) and imaged with an AxioObserver 7 (Carl Zeiss) wide-field microscope equipped with Plan Apochromat 20 × NA = 0.8 objective (Zeiss) and AxioCam 702 mono camera (Zeiss) and controlled by Zen 2.3 Blue (Zeiss) software. Image analysis was performed with Imaris 8.0.2 (Bitplane).

The co-localization scoring for MerTK and TIM4 with TUNEL was done with Imaris 8.2 (Bitplane). TUNEL$^+$ cells were detected with the Spots function, while MerTK$^+$ and TIM4$^+$ cells were detected with the Surface function. Spots that co-localize with Surfaces were identified with the 'Find Spots close to Surface' function of Imaris XT. The threshold for co-localization was set to 5 µm. The results were manually curated so that Spots categorized as 'not co-localized' that were: (1) at the edge of the imaging field were excluded from consideration; (2) with clear surface signal around them were re-categorized as 'co-localized'. The ratio of co-localized Spots to all Spots was calculated and presented as the co-localization index.

## Thymus transplantation

To obtain E15.5 embryos, *Spic*$^{GFP}$ (CD45.2) homozygous male and C57BL/6 (CD45.2) female mice were mated in a cage overnight and separated on the next day. Pregnant mice were sacrificed 15 days later, the viable embryos were harvested, and the thymuses were isolated in ice-cold PBS. Congenic CD45.1 recipients were anesthetized by i.p. injection of ketamine hydrochloride (120 µg/g, Toronto Research Chemicals) and xylazine hydrochloride (12 µg/g, Sigma). The fur on the left flank was removed, and the left kidney was exposed by cutting the skin, muscle layer, and peritoneum. The kidney capsule was nicked with a G23 needle, and the fetal thymus was pushed into the pocket under the kidney capsule with a G23 needle equipped with a plunger from a spinal needle. After the kidney was re-positioned back into the peritoneal cavity, the peritoneum was sutured, and the skin was stapled with metal clips. Rymadil (Carprofen, 5 µg/g, Zoetis) was given subcutaneously to ease the wound pain, and Trimerin (Sulfadiazine at 0.5 mg/mL + Trimethoprim at 0.1 mg/mL) was given in the drinking water for the first 2 weeks after the surgery. The metal clips were removed from the skin after the first week, and the transplanted thymus and recipient's endogenous thymus were harvested and analyzed 6 weeks after the kidney transplantation.

## Statistical analysis

Comparison between groups was made with Prism 6 (GraphPad Software). Comparisons between two groups were carried out with unpaired Student's t-test. When more than two groups were compared, a one-way ANOVA with Tukey correction was used. Differences were considered significant if $p < 0.05$.

## Acknowledgements

We are grateful to the IMMGEN consortium for performing the RNA sequencing of our samples and providing access to its database. We thank the following core facilities at National Yang Ming Chiao Tung University: Instrumentation Resource Center for providing access to its sorting and imaging facility and the Animal Facility for mouse husbandry. We are very grateful to Wan-Chun Chang, Bing-Xiu Guo, Chien-Yi Tung, and Kate Hua from the Genomics Center for Clinical and Biotechnological Applications of NCFB (NYCU, Taipei) for help with scRNA-Seq. Special thanks go to Dr. Fang Liao, Academia Sinica, Taipei, for help with cell sorting and for providing the 24G2 hybridoma, and Chang-Feng Chu for technical assistance. We are very grateful to Alexandra Norris, Cecilia Tai, and

Angelina Shih for the manual scoring of the in vitro phagocytosis experiments. We are indebted to Dr. Yu-Wen Su (NHRI, Taiwan) for help with the mice. Some figures were made with BioRender.

This work was supported by the Ministry of Science and Technology, Taiwan (grants # 107–2320-B-010–016 -MY3, 110–2320-B-A49A-521 -, and 111–2320-B-A49 –031 -MY3 to IL Dzhagalov), the Yen-Tjing Ling Medical Foundation (grant # CI-111–6 to IL Dzhagalov), and the Cancer Progression Research Center (NYCU). The authors declare no competing financial interests.

## Additional information

### Funding

| Funder | Grant reference number | Author |
|---|---|---|
| Ministry of Science and Technology, Taiwan | 107-2320-B-010 -016 -MY3 | Ivan L Dzhagalov |
| Ministry of Science and Technology, Taiwan | 110-2320-B-A49A-521 - | Ivan L Dzhagalov |
| Ministry of Science and Technology, Taiwan | 111-2320-B-A49 -031 -MY3 | Ivan L Dzhagalov |
| Yen Tjing Ling Medical Foundation | CI-111-6 | Ivan L Dzhagalov |

The funders had no role in study design, data collection and interpretation, or the decision to submit the work for publication.

### Author contributions

Tyng-An Zhou, Formal analysis, Investigation, Methodology, Writing – review and editing; Hsuan-Po Hsu, Chih-Yu Lin, Investigation, Methodology, Writing – review and editing; Yueh-Hua Tu, Hsuan-Cheng Huang, Formal analysis, Visualization, Writing – review and editing; Hui-Kuei Cheng, Investigation, Methodology; Nien-Jung Chen, Jin-Wu Tsai, Ellen A Robey, Resources, Writing – review and editing; Chia-Lin Hsu, Conceptualization, Resources, Supervision, Writing – review and editing; Ivan L Dzhagalov, Conceptualization, Data curation, Formal analysis, Supervision, Funding acquisition, Investigation, Visualization, Writing – original draft, Writing – review and editing

### Author ORCIDs

Tyng-An Zhou http://orcid.org/0000-0003-4031-4947
Ellen A Robey http://orcid.org/0000-0002-3630-5266
Ivan L Dzhagalov http://orcid.org/0000-0001-9209-4582

### Ethics

All experimental procedures involving animals were approved by the Institutional Animal Care and Use Committee (IACUC) of National Yang Ming Chiao Tung University (animal protocols #1070506, and 1090301). All surgery was performed under Ketamine + Xylazine anesthesia, and every effort was made to minimize suffering.

### Decision letter and Author response

Decision letter https://doi.org/10.7554/eLife.75148.sa1
Author response https://doi.org/10.7554/eLife.75148.sa2

## Additional files

### Supplementary files

• Transparent reporting form

### Data availability

The RNA Sequencing data of thymic macrophages and thymic dendritic cells are available at NCBI Gene Expression Omnibus (GEO) as part of GSE122108 and at https://www.immgen.org. The single

cell RNA sequencing data is deposited at NCBI GEO under accession number GSE185460. The source data underlying *Figure 1G and H*, *Figure 2*, *Figure 3B, D, and G*, *Figure 5C, F, and I*, *Figure 6B, E, G, and I*, *Figure 7B, C, D, and G*, *Figure 8B, D, E, and F*, *Figure 1—figure supplement 4*, *Figure 2—figure supplement 1*, *Figure 2—figure supplements 2 and 3*, *Figure 5—figure supplements 1 and 2* are provided in the Source Data files. All other data supporting the findings of this study are available within the article.

The following dataset was generated:

| Author(s) | Year | Dataset title | Dataset URL | Database and Identifier |
|---|---|---|---|---|
| Zhou TA, Dzhagalov IL | 2021 | Single-cell RNA-sequencing of thymic myeloid cells from Csf1rgfp/gfp (MaFIA) and Cd11cyfp/yfp mice | https://www.ncbi.nlm.nih.gov/geo/query/acc.cgi?acc=GSE185460 | NCBI Gene Expression Omnibus, GSE185460 |

The following previously published dataset was used:

| Author(s) | Year | Dataset title | Dataset URL | Database and Identifier |
|---|---|---|---|---|
| Immunological Genome Project Consortium | 2018 | OpenSource Mononuclear Phagocytes Project | https://www.ncbi.nlm.nih.gov/geo/query/acc.cgi?acc=GSE122108 | NCBI Gene Expression Omnibus, GSE122108 |

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
