## [Editor Report]

This work provides thorough characterization of thymic macrophages. The authors used bulk RNA-seq, single-cell RNA-seq and fate mapping animal models to demonstrate the phenotypes, origin and diversity of thymic macrophages. The manuscript is well written and the conclusions are mostly well supported by data.

---

## [Decision Letter]

**Decision letter after peer review:**

Thank you for submitting your article "Thymic macrophages consist of two populations with distinct localization and origin" for consideration by *eLife*. Your article has been reviewed by 3 peer reviewers, one of whom is a member of our Board of Reviewing Editors, and the evaluation has been overseen by Tadatsugu Taniguchi as the Senior Editor. The following individual involved in review of your submission has agreed to reveal their identity: Camille Bleriot (Reviewer #3).

Essential revisions:

1) The description of the gating strategy of thymic macrophages for Figure 1 is quite verbose. Adding a step-wise gating strategy of thymic macrophages as Figure 1a would be helpful for readers to follow the experimental details. Why not using the color map axis mode from FlowJo to show 3 parameters at a glance?

2) The authors should state what does row min row max in figure2 b,d refer to. Is this expression value on log scale? In figure 2d, the authors compared their own RNAseq data with ImmGen seq data. What kind of normalization did the authors apply?

3) The authors used immunofluorescence to identify the localization of two populations of macrophages, where they used MerTK staining to indicate all macrophages. However, MerTK expression may not restrict to immune cells. The authors are encouraged to confirm that MerTK only labels macrophages in thymus by co-staining with F4/80 or CD45. Tim4 can also be used in immunofluorescence.

4) About the plan, authors study the origin of the thymic population and provide data in Figure 2, 3 and 4 assuming that thymic macs form a homogeneous population. But from Figure 5, they distinguish 2 populations and study them separately. So the end of the paper renders obsolete the beginning, that asks for a revision of the whole plan.

5) The figure 2 could be revised also. First, the panel 2a is useless and should be removed. A PC analysis of all the macs would be more useful here. Also, the color code used for the genes is confusing. Why genes up in ThyMacs are red in 2b but only half of them in 2d? Info can be found in the legend but it should be more clear on a graphical point of view.

6) For figure 3, what is the time point of the panel 3b? Here, authors should show microglia and ThyMacs for both timepoints and conclude based on the comparison. If ThyMacs are as stable as the microglia, no replacement. If not, replacement. For the panel 3f, n=3 is too low to be convinced notably with the standard variation here. And displaying the dot plot with 11% of blood mono from donor while the median being around 20 is not fair, authors should present the most representative plot. For the panel 3h, there are more GFP (in term of MFI) for TEC and ThyMacs than for total cells. How is it possible? TECs and ThyMacs should be in the total cells? Or the gating is not clear enough?

7) For figure 4, the EdU staining (4e) is not convincing. The signal is very low (as compared to 4c for example.

8) For figure 7, the interpretation of the data and the way to present them are not clear. Authors use an inducible fate-mapping model. The fact that Tim4- loose their signal with time argue for a replacement by non-labelled cells (blood monocytes) whereas Tim4+ ones are stable meaning they self-maintain. It is what authors claim. But how it fits with previous data where they say that Tim4+ derived from CX3CR1+? The explanation that is a bit subtended here but not enough clearly shown is that CX3CR1+ give rise to Tim4+ during embryonic development but is stops after, Tim4 self-renew independently, and CX3CR1+ are slowly replaced by monocytes. As this is the central claim of the paper, it should be most clearly reported and for this, a substantial change of the whole plan is required.

*Reviewer #2 (Recommendations for the authors):*

1) In Abstract, the authors claimed that thymic macrophages "express typical tissue-resident macrophage genes", it is not clear what are these genes referred to.

2) It is an overstatement to say" Timd4+ macrophages occupied the cortex, while Cx3cr1+ cells were located in the medullar and the cortico-medullary junction" in discussion based on current IF data (first paragragh). The authors are encourage to perform additional staining to confirm their results.

*Reviewer #3 (Recommendations for the authors):*

Most of the data are here but the way to report them should be improved significantly.

[Editors’ note: further revisions were suggested prior to acceptance, as described below.]

Thank you for resubmitting your work entitled "Thymic macrophages consist of two populations with distinct localization and origin" for further consideration by *eLife*. Your revised article has been evaluated by Tadatsugu Taniguchi (Senior Editor) and a Reviewing Editor.

The manuscript has been improved but there are some remaining issues that need to be addressed, as outlined below in the reviewers' comments.

*Reviewer #2 (Recommendations for the authors):*

In the revised manuscript, the authors have provided additional data and a clearer explanation to address the reviewers' concerns. The authors have addressed all my issues with convincing data and proper discussion.

I have two more comments for the authors:

The authors showed two macrophage populations, yet in the EdU labeling experiment, they showed EdU staining in total macrophages. If the author can show EdU staining separately in Cx3cr1+ and Timd4+ macrophages, this data will be better to support their conclusion.

The authors show that the thymic macrophages are the dominant phagocyte in thymus and the IF data also suggested that Thy-macs effectively taking up apoptotic cells. Efferocytosis is related to the upregulation of many ani-inflammatory/pro-resolution genes in macrophages. Some studies have shown that efferocytosis promotes the TGFb1 production from macrophages, could that be related to the accumulation of ECM during thymus involution? Some discussion related to this point is encouraged.

*Reviewer #3 (Recommendations for the authors):*

The authors have substantially improved their manuscript and have taken into account the comments made by the referees after the first round. The manuscript reveals interesting features of thymic macrophages and deserves to be published.

---

## [Author Response]

Essential revisions:1) The description of the gating strategy of thymic macrophages for Figure 1 is quite verbose. Adding a step-wise gating strategy of thymic macrophages as Figure 1a would be helpful for readers to follow the experimental details. Why not using the color map axis mode from FlowJo to show 3 parameters at a glance?

We thank the reviewers for the suggestion. The description of the gating strategy has been stripped to 2 panels that capture its essence. At the reviewer’s suggestion, we also included a color map axis to show MerTK, CD64, and F4/80 in one plot.

2) The authors should state what does row min row max in figure2 b,d refer to. Is this expression value on log scale? In figure 2d, the authors compared their own RNAseq data with ImmGen seq data. What kind of normalization did the authors apply?

We apologize for not making this clear. The values in Figure 2b and d (current Figure 2A and C) are expression values on a log scale. We have included this information in the figure.

Our data is part of the IMMGEN dataset. We sorted the cells and sent them to the US for RNA sequencing. That is why we referred to them as “our” data. However, to avoid confusion, we changed the wording to clearly reflect that the data are from IMMGEN.

3) The authors used immunofluorescence to identify the localization of two populations of macrophages, where they used MerTK staining to indicate all macrophages. However, MerTK expression may not restrict to immune cells. The authors are encouraged to confirm that MerTK only labels macrophages in thymus by co-staining with F4/80 or CD45. Tim4 can also be used in immunofluorescence.

We agree that staining with additional macrophage markers will strengthen our conclusions about ThyMacs localization. We have performed staining with CD64 together with MerTK or Tim4. CD64 and MerTK almost completely overlapped, and so did CD64 and Tim4 in the cortex. We could not stain MerTK and Tim4 together because the antibodies are raised in the same species (rat). Additional evidence for the specificity of these markers for thymic macrophages comes from Figure 3E and F, showing the high degree of co-localization of apoptotic cells (TUNEL+) with MerTK or Tim4. Finally, Figure 4 figure supplement 1 also clearly shows the distribution of TIM4 and CD64 in the whole thymus.

4) About the plan, authors study the origin of the thymic population and provide data in Figure 2, 3 and 4 assuming that thymic macs form a homogeneous population. But from Figure 5, they distinguish 2 populations and study them separately. So the end of the paper renders obsolete the beginning, that asks for a revision of the whole plan.

We agree with the reviewer that there is more than one way to tell this story, and we have been agonizing over our plan. However, we respectfully disagree that the beginning of the paper is made obsolete by the ending for several reasons:

1) The initial figures in our manuscript contain a very fundamental characterization of ThyMacs. Just as the revelation of heterogeneity in liver macrophages (e.g., Sierro et al., 2017, MacParland et al., 2018) does not render all prior research on these cells obsolete, the initial figures in our manuscript are an essential part of the story. Such data are available for all other studied tissue-resident macrophage populations. Removing them will be a disservice to the community.

2) Another reviewer asked for a deeper characterization of ThyMacs based on the data in Figure 2. Accommodating this request will be very difficult if we remove this part.

Nevertheless, we agree that ThyMacs heterogeneity is the central claim of the manuscript and should be introduced earlier. The original figure 5 (current Figure 4) that describes the heterogeneity has been moved before the original figures 3 and 4 (current Figure 5 and 6). Additional analyses distinguishing Tim4+ and Tim4- ThyMacs have been incorporated in current Figure 5 and 6.

5) The figure 2 could be revised also. First, the panel 2a is useless and should be removed. A PC analysis of all the macs would be more useful here. Also, the color code used for the genes is confusing. Why genes up in ThyMacs are red in 2b but only half of them in 2d? Info can be found in the legend but it should be more clear on a graphical point of view.

We have revised Figure 2 according to the reviewer’s suggestions. The PCA analysis is consistent with the hierarchical clustering and shows that splenic and liver macrophages are most closely related to ThyMacs. We agree that the presence of red in both heatmaps is confusing, and we have changed the color code – color was removed from current Figure 2A but retained in Figure 2C.

6) For figure 3, what is the time point of the panel 3b? Here, authors should show microglia and ThyMacs for both timepoints and conclude based on the comparison. If ThyMacs are as stable as the microglia, no replacement. If not, replacement. For the panel 3f, n=3 is too low to be convinced notably with the standard variation here. And displaying the dot plot with 11% of blood mono from donor while the median being around 20 is not fair, authors should present the most representative plot. For the panel 3h, there are more GFP (in term of MFI) for TEC and ThyMacs than for total cells. How is it possible? TECs and ThyMacs should be in the total cells? Or the gating is not clear enough?

We thank the reviewer for pointing out our omissions. Figure 3b (current Figure 5B) is from E19.5, and we have added this information to the figure. We also agree that in Figure 3f (current Figure 5F), the sample number is too small and the variation too large to make solid conclusions. That is why we have repeated the partial chimeras experiment trying to irradiate as much as possible of the mice without affecting the thymus. We have substituted the data in Figure 3e and 3f with the new data. For Figure 3h, we apologize for not labeling the data clearly. The panels labeled “single, live cells” should be marked as “thymocytes” as they were obtained without enzymatic digestion, which is essential for both TECs and ThyMacs. However, we found an important caveat in the thymus transplant experiment. Some of the thymus macrophages appeared GFP positive not because they express GFP but because they have engulfed GFP+ cells. As a result, our experiments with embryonic GFP+ thymus transplants overestimate the percentage of donor-derived ThyMacs (all of them were GFP+). We have repeated the thymus transplantation experiments with congenically marked thymuses (CD45.2 donor and CD45.1 host). While this setup did not allow us to use the thymic epithelial cells as a positive control because they are CD45-, we did identify host-derived ThyMacs, consistent with Tim4- cells originating from adult HSCs. Thus, we have replaced the previous data in Figure 3H and 3I with current figures 5H and 5I.

7) For figure 4, the EdU staining (4e) is not convincing. The signal is very low (as compared to 4c for example.

We agree that the signal after 21d chase is much weaker than after 2 h (Figure 4c) or 21d (Figure 4e) of EdU pulse. The reason we decided to keep this data is that: (1) the thymocytes also have much lower EdU staining after 21d chase compared to 2h and 21d of EdU pulse; (2) The results from EdU staining are very consistent with the data from Ki67 staining, cell cycle analysis, and scRNA-Seq revealing a small population (~5%) of cycling ThyMacs.

8) For figure 7, the interpretation of the data and the way to present them are not clear. Authors use an inducible fate-mapping model. The fact that Tim4- loose their signal with time argue for a replacement by non-labelled cells (blood monocytes) whereas Tim4+ ones are stable meaning they self-maintain. It is what authors claim. But how it fits with previous data where they say that Tim4+ derived from CX3CR1+? The explanation that is a bit subtended here but not enough clearly shown is that CX3CR1+ give rise to Tim4+ during embryonic development but is stops after, Tim4 self-renew independently, and CX3CR1+ are slowly replaced by monocytes. As this is the central claim of the paper, it should be most clearly reported and for this, a substantial change of the whole plan is required.

We thank the reviewer for pointing out the need for a better explanation. The maintenance of the different populations of ThyMacs is indeed complex and proceeds in different ways in different periods of life. We have added some extra data to Figure 7 (current Figure 8) that we hope will add some clarity to the maintenance of thymic macrophages with age. The new Figure 8F shows the dynamics of the cell numbers of Tim4+ and Tim4- macrophages with age. Tim4+ cells peak in young mice and decline significantly as mice age. So, we do not think they are self-maintaining but, undergo slow attrition with minimal replacement. These results are consistent with Figure 6I showing low levels of Mki67 in Tim4+ cells. Tim4- cells are a different story: they progressively accumulate with age. Although the variability in thymus size and Tim4- macrophages in very old mice is too great for the data to reach significance, the trend is clear.

As for the dynamics of the populations in the embryonic period, we added data formally demonstrating that TIM4+CX3XR1- are derived from CX3CR1+ cells by fate mapping (Figure 7E-G). We induced re-combination in pregnant ROSA26LSL-GFP mice pregnant from Cx3cr1CreER males at E15.5 when almost all ThyMacs are Cx3cr1+ (Figure 7A). Just before birth, at E19.5, we could find a substantial proportion of TIM4+CX3CR1- cells among the fate mapped GFP+ macrophages, indicating that Cx3cr1+ cells, indeed, give rise to TIM4+CX3CR1- cells. As pointed out before, this pathway gets exhausted by the first week after birth – at d7, all ThyMacs are TIM4+.

Reviewer #2 (Recommendations for the authors):1) In Abstract, the authors claimed that thymic macrophages "express typical tissue-resident macrophage genes", it is not clear what are these genes referred to.

In the initial version of the manuscript the “typical tissue-resident macrophage genes” referred to the genes in Figure 2A and B. This list was taken from the original characterization of tissue resident macrophages by IMMGEN in Gautier et al. Nat Immunol 2012 Table 1. These genes are expressed in all tissue-resident macrophage populations studied but not in dendritic cells. However, we realized the vagueness of this statement and as another reviewer objected to Figure 2A, we decided to omit it from the Abstract.

2) It is an overstatement to say" Timd4+ macrophages occupied the cortex, while Cx3cr1+ cells were located in the medullar and the cortico-medullary junction" in discussion based on current IF data (first paragragh). The authors are encourage to perform additional staining to confirm their results.

We thank the reviewer for the suggestion to confirm the results of our initial staining with direct staining for TIM4. As shown in Figure 4 figure supplement 1, TIM4 staining is indeed restricted to the cortex and virtually all CD64 macrophages in the cortex express TIM4. In contrast, in the medulla, CD64+ macrophages do not express TIM4. The combined Figure 4 and Figure 4 figure supplement 1 substantiate our initial statement.

[Editors’ note: further revisions were suggested prior to acceptance, as described below.]

The manuscript has been improved but there are some remaining issues that need to be addressed, as outlined below in the reviewers' comments.Reviewer #2 (Recommendations for the authors):In the revised manuscript, the authors have provided additional data and a clearer explanation to address the reviewers' concerns. The authors have addressed all my issues with convincing data and proper discussion.I have two more comments for the authors:The authors showed two macrophage populations, yet in the EdU labeling experiment, they showed EdU staining in total macrophages. If the author can show EdU staining separately in Cx3cr1+ and Timd4+ macrophages, this data will be better to support their conclusion.

We agree with the reviewer that EdU staining of Cx3cr1+ and Timd4+ macrophages will strengthen the conclusion that the Cx3cr1+ subset is more proliferative that is currently based only on scRNA-Seq data. However, this experiment is going to take at least 1.5 months to complete. We would offer instead, staining for Ki67 that shows that, indeed, TIM4- thymic macrophages express higher protein levels of Ki67 than TIM4+ confirming that they are more proliferative (Figure 6I). We hope the reviewer will find this experiment acceptable.

The authors show that the thymic macrophages are the dominant phagocyte in thymus and the IF data also suggested that Thy-macs effectively taking up apoptotic cells. Efferocytosis is related to the upregulation of many ani-inflammatory/pro-resolution genes in macrophages. Some studies have shown that efferocytosis promotes the TGFb1 production from macrophages, could that be related to the accumulation of ECM during thymus involution? Some discussion related to this point is encouraged.

The reviewer is raising an important question about the role of efferocytosis-induced TGFb production for ECM accumulation and thymic involution. Unfortunately, the data we have do not allow a straight-forward answer to this question. The thymic macrophages, indeed, have the highest expression of TGFb in our scRNA-Seq data. However, the Timd4+ subset expresses significantly higher levels of Tgfb1 than the Cx3cr1+ subset. This fact casts some doubt that Tgfb1 is the primary driver for ECM accumulation during thymic involution because the peak in the number of Timd4+ macrophages is in young mice (Figure 8F). At that time there is minimal ECM in the cortex where these cells reside. In addition, during thymic involution the number of these cells declines significantly. The clear correlation between the accumulation of Cx3cr1+ thymic macrophages and thymic involution suggests that some factor(s) produced exclusively by these cells, such as PDGFa, will be more relevant. We have this information to the Discussion (lines 824-835)